# The Complex Biological Effects of Pectin: Galectin-3 Targeting as Potential Human Health Improvement?

**DOI:** 10.3390/biom12020289

**Published:** 2022-02-10

**Authors:** Lucas de Freitas Pedrosa, Avraham Raz, João Paulo Fabi

**Affiliations:** 1Department of Food Science and Experimental Nutrition, School of Pharmaceutical Sciences, University of São Paulo, São Paulo 05508000, SP, Brazil; lfpedrosa@usp.br; 2Department of Oncology and Pathology, School of Medicine, Karmanos Cancer Institute, Wayne State University, Detroit, MI 48201, USA; raza@karmanos.org; 3Food and Nutrition Research Center (NAPAN), University of São Paulo, São Paulo 05508080, SP, Brazil; 4Food Research Center (FoRC), CEPID-FAPESP (Research, Innovation and Dissemination Centers, São Paulo Research Foundation), São Paulo 05508080, SP, Brazil

**Keywords:** galectin-3, pectin, structure and function, bioactive polysaccharides, galectin-3 inhibition

## Abstract

Galectin-3 is the only chimeric representative of the galectin family. Although galectin-3 has ubiquitous regulatory and physiological effects, there is a great number of pathological environments where galectin-3 cooperatively participates. Pectin is composed of different chemical structures, such as homogalacturonans, rhamnogalacturonans, and side chains. The study of pectin’s major structural aspects is fundamental to predicting the impact of pectin on human health, especially regarding distinct molecular modulation. One of the explored pectin’s biological activities is the possible galectin-3 protein regulation. The present review focuses on revealing the structure/function relationship of pectins, their fragments, and their biological effects. The discussion highlighted by this review shows different effects described within in vitro and in vivo experimental models, with interesting and sometimes contradictory results, especially regarding galectin-3 interaction. The review demonstrates that pectins are promissory food-derived molecules for different bioactive functions. However, galectin-3 inhibition by pectin had been stated in literature before, although it is not a fully understood, experimentally convincing, and commonly agreed issue. It is demonstrated that more studies focusing on structural analysis and its relation to the observed beneficial effects, as well as substantial propositions of cause and effect alongside robust data, are needed for different pectin molecules’ interactions with galectin-3.

## 1. Introduction

Pectins are complex polysaccharides and versatile hydrocolloids, vastly available in plant cell walls and the middle lamella of higher plants. These polysaccharides are major components for maintaining rigidity and integrity of the plant tissues, positively impacting plant growth and health [1]. Every source of pectin has a variable amount of sub-structures, such as homogalacturonans (HG), rhamnogalacturonans (RG-I and II), and xylogalacturonan [2]. The HG region is primarily composed of a homopolymer of partially esterified 1-4-α-d-galactopyranuronic acid (Gal*p*A). RG-I regions are composed of repeated and intercalated α-d-Gal*p*A with α-l-rhamnopyranose (Rha*p*) [→2)-α-l-Rha*p*-(1,4)-α-d-Gal*p*A-(1→]. This region also has variable side chains of galactans (β-1,4-Gal*p* residues with varying degree of polymerization), arabinans (α-1,5-l-Ara*f* with 2- and 3- linked arabinose/arabinan branches), and arabinogalactans (Type I: β-1,4-d-galactans with *O*-3 linked l-arabinose or arabinan; Type II: β-1,3-linked d-galactans with β-*O*-6-linked galactans or arabinogalactan), attached to the Rha*p* residues (side chains are linked at Rha*p O*-4, as demonstrated in Figure 1 and Figure 2) [2,3,4]. In diverse fruits, these structures undergo a series of chemical and enzymatic alterations during the ripening process, resulting in a wide variety of intramolecular changes inside the pectic chain, as illustrated before for the papaya pulp, a fruit that rapidly undergoes the ripening process [5,6,7,8].

As part of the dietary fiber group, pectins are not digested by the human tract, although the fermentation process by the human microbiota is greatly important for the maintenance of the colonic and systemic health through local signaling. The fermentation products and molecular fragments can improve metabolic syndrome and attenuate hypercholesterolemia, hypertriglyceridemia, and hyperglycemia, markers related to heart disease risks in mice, rats, and humans [10,11,12,13,14]. The pectic products and fragments after colonic fermentation also possess anti-oncogenic attributions in the colon for mitigating cancer-related risks in the colon and even in other types of neoplasia in humans [15].

Galectin-3 (Gal-3) is the chimeric representative member of the galectin protein family, located ubiquitously in the nucleus, cytoplasm, outer cell surface, and extracellular space in mammals. Gal-3 is classified as a β-galactoside binding protein, although—as it is discussed further ahead—its binding range cannot be restrained only to β-galactoside ligands. Gal-3 is composed of a flexible N-terminal domain, containing up to 150 amino acid residues with sequences rich in proline, tyrosine, glycine, and glutamine. This collagen-like region rich in proline, glycine, and tyrosine ends up in a C-terminal domain with a carbohydrate recognition domain (CRD) containing about 135 amino acids. The CRD region is responsible for the signature pattern of the galectins family [16,17,18]. Although the ubiquitous expression, the main biological source is derived from immunological and collagen-producing cells, where it helps to establish cell recognition and communication through protein–protein or glycan–protein interactions [19,20]. For example, one important and recently elucidated Gal-3 effect in physiological conditions is the recruitment of endosomal sorting complexes required for transport to damaged lysosomes (ESCRTs), so they can be effectively repaired [21]. However, alterations of Gal-3 expression are strongly related to tumor growth, cancer cell proliferation, cell-to-cell adhesion properties, fibrosis stimulation, T-lymphocytes apoptosis, macrophage differentiation into infiltrative forms that stabilize tumor environment, and other features [22,23,24,25,26,27,28]. Those attributions are shown in Figure 3, where the intestinal model was chosen, as it is simple to explain the interface between endogenous and exogenous stimuli that differently activate Gal-3 functions. Moreover, pectin is considered a bioactive compound when ingested as a food component or as a dietary supplement, a topic that is explained in the next sections. It can be noted in Figure 3 the different Gal-3 forms available at biological environments, such as the monomeric unit (intra and extracellular) recently secreted present at initial interactions with natural ligands and pentameric (chimeric appearance) after associations of other monomeric units through their N-terminal domain due to ligand stimulus [18,29].

While there are extensive high-quality data regarding Gal-3 binding sites modeled to small glycoconjugates and an important level of binding/inhibition by those [30,31], the present review looks forward to establishing ways described in the literature regarding pectin and its fragments and different molecular interactions with Gal-3 protein, as it is highly controversial due to structural variability and reproducibility of experimental conditions. This review also covers the Gal-3 independent pectin interactions. The gathering of these data is fundamental for a better understanding of the overall perspectives regarding this promising—but not yet fully understood—area to be explored. Likewise, all the structural parameters, such as distinct molecular sizes, a higher number of galactan/arabinan side chains or galacturonic acid contents, methyl esterification, and other properties important to pectin biological effects are also discussed by in vitro and in vivo perspectives [7,32,33].

## 2. Basic Pectin Molecular Aspects

### 2.1. Pectin Molecular Weight

Although having extremely variable structures, pectins are composed of large polysaccharide chains translated to solutions with high viscosity, a factor which would likely be an impairing factor for possible absorption through oral administration [34]. There are several approaches for extracting and, at the same time, lowering pectic molecular weight. The most commonly utilized method, especially in industrial extraction, is the chemical modification with mineral acids [35], such as hydrochloric and nitric acids, and organic acids [36,37,38,39], such as citric and malic acids. Enzymatic digestion is a more subtle and precise approach, using different cleavage agents, mostly arabinanases and galactanases, to remove excess side chains and polygalacturonases (exo- and endo-) to break down the larger HG backbones/pectic domain, aiming to obtain pre-planned molecular patterns [33,40]. Another category, thermic modifications, is less specific but far more practical and inexpensive. These modifications can be achieved by different methods, such as high temperature and pressure [41,42], ultrasound-assisted heating [43,44,45], and more underused but promising approaches, such as electromagnetic induction heating [46]. There are described extraction methods capable of conserving more of the native pectin structure, such as the use of low temperatures (40 °C) but with low extraction yields, and the decision which technique should be used depends on the final objective of the work [47], as it is discussed further that high molecular weight polysaccharides are not well suited for potential biological applications. Therefore, breaking down and/or manipulating the large polysaccharide chains into smaller portions while bringing better malleability to the molecule itself, improving viscosity and rheological properties, can set more biologically available binding sites, facilitating the potential interaction with proteins such as lectins or other cell-surface receptors.

### 2.2. Monosaccharides, Backbone, and Side Chains

The monomeric structure of polysaccharides is also extremely variable and inconsistent depending on the food source, extraction method, and modification strategy. These monomeric sugars are naturally presented as either pyranose (*p*, saccharides with chemical structure including six-member ring, composed of five carbon and one oxygen atoms) or furanose (*f*, saccharides with chemical structure including five-member ring, composed of four carbon and one oxygen atoms). This high variability represents a problem regarding standardization of recommended ratios between monosaccharides, such as the Gal*p*/Ara*f*, Ara*f* + Gal*p*/Gal*p*A, associated with the proportions of molecular side chains (e.g., arabinans and galactans) or main structure sequences of α-d-Gal*p*A. For example, important works regarding molecular modeling of Gal-3 inhibitors that have standardized to low molecular weight molecules (<1000 Da) as higher protein-inhibitor ligands [48,49,50] did not consider polysaccharides. Galactan and galactosyl residues were the main focus as contributors to biological effects observed between pectin and Gal-3 interaction and inhibition since this protein has a preference for binding β-d-galactopyranoside [40,51]. However, while still important targets of interest, other monomeric compounds such as the Gal*p*A, Ara*f*, and their association have been described more consistently, as found for polysaccharides with Ara*f* residues in higher quantities, mostly linked in α-(1→3,5)-l-arabinan side chains, demonstrating higher inhibition of Gal-3 through techniques such as Gal-3 hemagglutination assays (G3H) or binding through biolayer interferometry assays (BLI) and surface plasmon resonance assays (SPR) [3,33,52,53]. Therefore, this focus on specific monosaccharides that are not only galactosyl could help to understand the biological action of pectin and the possible Gal-3 interaction and inhibition.

### 2.3. Esterification Degree

Pectin can also be classified by the esterification degree throughout its molecule, where the common approach is the division in a lower degree of esterification (DE) (<50% DE—low-methylation or LMP) or higher DE (>50%—high-methylation or HMP), as well as how the esterification is distributed on pectin molecule (degree of blockiness). The determination of this parameter as a characterization step is crucial to establish the best application of the referred polysaccharide, such as the gelling property and potential where HMP achieves through hydrophobic interactions and hydrogen bonding, while LMP forms gels through the salt-bridge connecting adjacent or opposite carboxyl groups from divalent ions [54,55], emulsion property for protein complexation or drug delivering [56,57,58], in addition to the direct biologic relationship that is further discussed in the following topics. The RG-I backbone region’s side chains are mostly a mixture of arabinans, galactans, and arabinogalactans attached to the rhamnose residue. RG-II is highly methyl-esterified and much more diverse in sugar composition side chains than RG-I. The main pectin backbone is composed of HG, also known as the “smooth” region. This main portion of pectic molecules can also exhibit varying degrees of methylation or acetylation (DM and DA, respectively) depending on the food source and extraction process [2]. Alkali treatment is the most common chemical method to achieve lower DM for pectic samples through direct saponification but may result in chemical waste or even slight alterations to the main pectin chain by β-elimination. Enzymatic treatment is less practical, requiring up to 25 h, depending on the expected DM. An alternative to this inconvenience is the high hydrostatic pressure-assisted enzymatic process, which is a promising operation to help attenuate this problem and facilitate the industrial application of this method [59].

### 2.4. Rheological Properties

When studying pectin rheology parameters, they are especially useful to determine potential applications of the characterized samples towards large-scale applications, both at food and non-food products, also helping at the definition of a better biologic destination [60]. The previously mentioned structural parameters directly impact the food product incorporation, such as viscoelasticity and thickness, which can be interpreted as gelling capacity, stabilizing potential in acidified milk beverages or fruit juices, emulsion capability with protein-rich solutions, and many others [61,62,63,64]. Therefore, the stratification generated from this type of analysis can lead to the best type of use for each pectin sample [65].

### 2.5. Food Source

The food source—exclusive from plants—together with the pectin extraction methods, and in some cases, the ripening parameters of fruits, are important factors in obtaining functional pectic molecules, as already mentioned. Usually, there is an additional ecologically sustainable status, as many of the possible sources are residues and byproducts from the juice and food processing industries. Some residues of apple [66], *Prunus domestica* and *Prunus mume* [67,68], jaboticaba [69], citrus [70,71], and papaya [7,8,72,73] have been explored for pectin extraction and biological activity studies.

## 3. Gal-3 Binding Sites and Pectin Interactions

Structurally, Gal-3 N-terminal tail (NT) transiently interacts with its CRD F-face and is linked to the glycine/proline-rich sequence, conferring its uniqueness in the galectin family, allowing self-oligomerization [74,75,76]. Gal-3 CRD is composed of 11 β-sheets, whereas five belong to the F-face and six belong to the conventional β-galactosides binding S-face, one opposed to the other, forming a β-sheet-sandwich (Figure 4A–C). In the S-face CRD, there is the NWGR conserved motif (Asparagine–Tryptophan–Glycine–Arginine), which is similarly encountered in the BH1 domain from B-cell Lymphoma-2 (Bcl-2) anti-apoptotic molecule, and suggested as the possible interaction that enables apoptosis evasion in tumoral cells (Figure 4B) [19,77,78]. Within the CRD, specifically the S-face, there are some subsites that can be named for easier comprehension of binding interactions. There are two conserved subsites (C and D), two non-conserved (A and B), and one not well-defined subsite (E) [30], where their interfaces are mainly constituted of hydrogen bonds. The CD-associated subsites are a target of natural ligands such as the N-acetyllactosamine (LacNAc) Gal-3 inhibitor or the β-d-galactopyranoside residue located in the Lactose molecule, where it remains tightly bonded mainly to C subsite amino acids (Figure 4D) [79].

However, there is extensive literature regarding other synthetic and sugar-derived molecules (especially tetrasaccharides and thiosaccharides) that utilize the whole ABCD region (also mentioned as Gal-3 binding groove, as its morphological 3D structure conformation due to the interactions between the groups AB/CD and each one of them individually) for a better affinity performance and inhibition potential [30,49,74,82]. In addition to this, the interaction of pectic poly- and oligosaccharides with Gal-3 is suggested through N-tail epitope recognition by pectin side-chain residues (e.g., galactans). The F-face interacts with β-galactosides (but also other portions), disrupting the CRD F-face binding with NT. Additionally, the same pectic structure could show multiple interactions involving both the S face region (disturbance of amino acids residues from the canonical binding site, e.g., 154–176 sequence observed with Heteronuclear single quantum coherence spectroscopy (HSQC spectra)) and F-face region (amino acids residues of β-sheets 7, 8, and 9 mostly, e.g., 210–225 sequence) (Figure 5) [76,83,84].

Other Gal-3 inhibitors that have been thoroughly and increasingly studied are pectin and its fragments. Some of the possible Gal-3 inhibition effects by pectin and fragments might include protecting pancreatic β-cells against oxidative and inflammatory stress [85]. Xu et al. [86] observed modified citrus pectin (MCP) downregulated pathways involved in myocardial fibrosis, but the authors did not study the Gal-3 inhibition by MCP in vivo, although Gal-3 was downregulated in the treated group [86]. Some other biological effects from Gal-3 inhibition are related to cancer proliferation control [7,32], with a strong positive relationship between overexpression of Gal-3 and carcinogenic processes, such as apoptosis evasion, higher cell invasion, and metastatic progression, which are key signatures in tumor and metastatic types of cells [87]. The structural relationship between the pectic chain and Gal-3 binding sites is also very important for the expected positive functional effects [33,88]. Extracellular Gal-3 biological functions are exerted mostly by interacting with glycoconjugates in cell surfaces, such as laminin and adhesin, signaling and activating specific pathways. Some studies regarding binding interactions between pectin and its fragments with Gal-3 are listed below (Table 1).

The Gal-3 binding range is considerably wide, including oligolactosamines, sulfated and sialylated glycans, and α-binding glycans, such as Fucα1-2, Galα1-3, and Galα1-4 added to the core of LacNAc molecule (as terminations) [89,90,93,94], which could indicate why non-β-Gal pectic fragments and other sources can still perform positively in some binding experiments. The β-1-4-galactan side chains inside the pectic RG-I domains have been highly attributed as the molecular factor involved in direct inhibition of Gal-3 (recombinant proteins and native proteins from cells), mainly discussed as a motif varying reaction, such as galactose residues in the middle or at terminal parts of those linear molecules. The longer galactan side chains in RG-I molecules identified in MCP and ginseng pectin were also associated with a stronger interaction with Gal-3 in vitro [40,95]. Similar positive binding affinity results with Gal-3, higher than observed with potato galactan, were obtained when isolating a 22.6 kDa RG-I polysaccharide from pumpkin [91]. Meanwhile, the RG-II enriched fractions extracted from *Panax ginseng* flower buds, with a high methyl-esterification and backbone substitution, as well as with lower content of galactose residues and low molecular weight, were associated with an absence of Gal-3 binding [3]. Highly esterified HG samples also did not inhibit Gal-3 [3]. However, other ginseng samples with an equivalent ratio between HG and RG-II regions had a positive binding affinity to Gal-3 in a similar way to the synergistic effect of HG and RG-I of citrus pectin [34,83]. In the latter case, the authors also suggested that the unesterified characteristic of pectin, alongside non-substituted GalA segments, were crucial structural elements for the observed biological effects [83]. Following this trend, commercial lemon pectin samples (from CP Kelco) with a low degree of methyl-esterification and low molecular weight were consistently and extremely potent in preventing negative outcomes in human islets with β-cell apoptosis (diabetes model) induced stress in vitro [85], in a dose–response manner. Gal-3 inflammatory stress induction was analyzed by evaluating oxygen consumption rate (OCR) in the presence and absence of a Gal-3 known inhibitor, α-lactose. After the previous incubation with α-lactose, the pectin sample effects that reversed OCR reduction induced by streptozocin and minimized inflammation induced by cytokine incubation were greatly compromised. The author’s suggestions were based on the observed effects derived from pectin binding to Gal-3 [85]. However, specific data regarding the connection between the in vitro effects and Gal-3 inhibition by pectins were not proposed.

As already mentioned before in this topic, Gal-3 does not have only one direct binding site toward polysaccharides, known as the canonical S-face (sugar-binding) region inside the CRD. This can help to explain the different results in distinct binding experiments based upon competitive inhibition and chain length [88]. Detailed data suggest that the Gal-3 N-terminal binds to a 60 kDa RG-I-rich pectin portion of ginseng through galactans located in RG-I ramifications [92]. This same molecule with removed ramifications (Rha*p* and Gal*p*A intercalated residues only) did not bind with Gal-3 [92]. Another molecule rich in galactose—but not derived from pectin—is a galactomannan isolated from *Cyamopsis tetragonoloba* guar gum flour (1-4-β-d-mannopyranose backbone with 1-6-α-d-galactose ramifications) also did not bind with Gal-3 [92]. However, experiments performed with potato galactans oligosaccharides derived from pectin had similar results as observed for the ginseng RG-I regarding Gal-3 binding [92], demonstrating the probable uniqueness structural features of pectin-derived molecules that can result in Gal-3 binding.

Gal-3 F-face has shown strong binding signals with pectin molecules, mainly at lower polysaccharide concentrations, opening more space to conventional S-face binding in higher concentrations. Like S-face, the F-face non-orthodox site is enriched by hydrophilic and charged amino acids residues. In addition to not having the Trp key-residue, other hydrophobic side chains could have similar functionality, as well as similar concave shape conformation [84,92] (Figure 4A–D). It is specified that resonance broadening was attributed as the primary capability of binding between Gal-3 and the RG-I-4 (a ginseng-derived rhamnogalacturonan), but also that broadening would be directly correlated to higher noise and lower sensitivity in analytical techniques [84,92], a factor which has to be taken into account. The presence of those non-conventional sites was suggested when lactose (CRD S-face inhibitor) did not interfere in RG-I binding to Gal-3 [40]. Furthermore, the binding of a determined ligand to the F-face of Gal-3 CRD could allosterically modify S-face residues. These shifts influence the conformation of the opposing face affecting the affinity between ligand (peptide, glycoconjugate, or polysaccharide) and the Gal-3, improving or attenuating the inhibitory/activity effects [76]. Moreover, Gal-3 N-terminal tail phosphorylation, although having little impact in CRD F-face, may be related to allosterically influencing carbohydrate-binding to the canonical CRD S-face site [76]. This structure-related information and respective identification method improvement contributed to more profoundly describing how the synergistic effects between pectin fragments work, as will be thoroughly discussed below.

Unusual non-galactoside poly-oligomeric ligands could act through multivalence interaction with Gal-3. The synthesis or identification of molecules that represent potential inhibitors of multimeric (chimeric) Gal-3 to cell-membrane glycoconjugates could be critical for minimizing known observed effects, such as the glycoclustering of receptors resulting in apoptosis induction of T cells [96]. Chelation, subsite binding, and reassociation of the binding site towards different monomeric structures within a molecule are also candidates for multivalent interactions with Gal-3 [97]. Although pectins and other natural ligands are harder to validate in comparison to synthetic compounds, parameters such as molecular flexibility and occupation of binding sites can be less manipulated/predicted [98]; the study of monosaccharide and consequently substructures ratios influencing Gal-3 binding is a promising area of investigation.

A recent in vitro study analyzed Gal-3 inhibition and MCF cell viability after specific enzymatic modification of citrus water-soluble fraction (WSF) rich in pectin. It was demonstrated that although β-1,4-galactan side chains in RG-I are still considered the most accountable molecular part for the observed effects, the partial removal of the side chains composed of other monosaccharide residues, such as α-1,5-arabinan, also contributed to inferior results both at cancer cell proliferation and direct gal-3 binding [33]. Other complex structures, such as the pectic acidic fractions extracted from *Camellia japonica* pollen, described as an RG-I-like polysaccharide, had their branched α-1-3,5 arabinan and type II arabinogalactans attributed to their strong Gal-3 inhibition effects [53]. It is suggested that, despite not having a significant relationship to the ratio of RG-I/HG, pectin-rich WSF bioactivity was dependent on cooperation between RG-I and HG regions [33]. In addition to this pectin fragment ratio, the monosaccharide residue composition ratio had an impactful performance. The Gal*p* amount is necessary for the best Gal-3 inhibitory results since pectic fragments with lower Gal*p*/*Araf* ratio resulted in lower Gal-3 inhibition even with similar molecular sizes than other studies (50 to 60 kDa) [52].

There are also in vitro data for anti-proliferative characteristics for sugar-beet pectin, in which both the RGI/HG backbone and the galactans/arabinan side chains exhibited those positive effects in HT-29 (human colorectal adenocarcinoma) cell populations [99]. The alkali treatment not only increased the RG-I/HG ratio (which translates to more neutral sugar side chains) but also enhanced the anti-proliferative effects. The removal of almost all side chains in the sample did not completely abolish the effects, denoting the importance of the cooperative effects between different pectic structures [99]. Similar results were obtained in our lab, in which papaya uronic fraction, a Gal*p*A fraction enriched with galactans, had the best in vitro results in inhibiting Gal-3 hemagglutination than neutral sugar-enriched fractions [7]. The difference of the samples was the degree of methylation, in which the former had low DM, and the latter had high DM.

Zhang and colleagues [83] indicated that a combination of RG and HG polysaccharides enhances the Gal-3 binding in vitro through HG interaction with RG, opening more Gal-3 binding epitopes in the RG molecule. At a particular RG/HG molar ratio, the interaction between polysaccharide and the F-face of Gal-3 could establish the new activated binding epitopes due to the higher prevalence of galactose residues, facilitating the S-face CRD interaction. This interaction between HG and RG could perform alterations in its conformation and increase or enhance the synergistic functional effects [83]. Each 130 kDa molecule of the isolated pectin fraction could bind up to 16 Gal-3 molecules. Overall, the combination of both structures showed biologically better activity than separated molecules [83]. Additional ginseng HG-rich fractions were responsible for inducing apoptotic process in vitro at higher doses and cell cycle arrest at lower doses in HT-29 cells, and these biological effects were increased after heat treatment of the polysaccharide fractions [100].

Another ginseng polysaccharide fraction had 91 kDa, an Ara*f*/Rha*p* ratio of 2:1, and a Gal*p*/Xyl*p* ratio of 1:1 and was characterized as xylo/rhamnogalacturonan I with arabinan/galactan side chains. This fraction was evaluated for in vitro Gal-3 inhibition, anticancer effect, and in vivo gut microbiota modulation [101]. The authors indicated that the xylans were mostly responsible for the microbiota’s healthy recovery and protection. Meanwhile, arabinogalactan side chains can interact with Gal-3 down-regulating tyrosinase through its N-glycan binding site [101]. Overall, the polysaccharide was effective in restoring normal levels of the important interleukin for tumor rejection and T cell activation, such as IL-10. The polysaccharide also modulated IL-17, a mediating molecule over-produced by the tumor cell microenvironment, revealing a multi-targeted functionality of the ginseng polysaccharide [101]. Although studied for many years, this pectic polysaccharide immunomodulatory property is still a trending topic, especially when the structural differences influence the interaction between the pectin and immune receptors, such as the toll-like receptor family (TLR) and interleukins at the macrophage cell surface [73,102], and will be more profoundly discussed later.

After determining the direct inhibition potential by competing with Gal-3 ligands in ECM, it was demonstrated that MCP can also downregulate Gal-3 expression. The in vitro induction of cell cycle arrest at the G2/M phase, through Cyclin B1 decrease and cyclin-dependent kinase 1 (Cdc2) phosphorylation, disrupts the Gal-3 pathological function of maintaining cell cycle arrest at the late G1 phase that leads to evasion from apoptosis induction [103]. Gal-3 can also induce phosphorylation in the signal transducer and activator of transcription 3 (STAT-3) in ovarian cancer cell spheroids. Although demonstrating slight decreases in cell viability after treating with paclitaxel, a strong synergistic effect between this drug and MCP was observed. The IC50 values decreased when both compounds were together, alongside a 70% increase in caspase-3 activity and a 75% Cyclin D1 expression level decrease against the values obtained using only paclitaxel [104].

Regarding fibrosis induction, MCP alleviated liver fibrosis and stress-induced secretions, such as decreased malondialdehyde (MDA), TIMP metallopeptidase inhibitor 1 (TIMP-1), collagen-1 α-1 (Col1A1), and Gal-3 expressions, improving HSC apoptosis rate and the upregulation of glutathione and superoxide dismutase in vivo [105]. Renal fibrosis-related biomarkers in an adult male Wistar murine model were also attenuated by treatment with MCP in drinking water. Albuminuria, proinflammatory cytokines, such as small inducible cytokine A2, osteopontin, epithelial transforming growth factor-β1, and other epithelial to mesenchymal transition factors, were all controlled or restored to normal levels after treatment in normotensive experimental models of renal damage [106].

Gal-3 is up-regulated by aldosterone, mediating inflammatory and fibrotic response in vascular muscle cells in vitro and in vivo [107]. The mechanism was demonstrated as inducing Gal-3 secretion by macrophages via phosphatidylinositol 3-kinase inhibitor/AKT and nuclear factor κB transcription signaling pathways in vitro and in vivo [108]. Furthermore, in aldosterone-induced cardiac and renal injuries, MCP subcutaneous injection in mice—different from the major in vivo experiments when MCP was diluted in drinking water—downregulated Gal-3 at protein and mRNA levels while also minimizing cardiac adverse effects induced by aldosterone salt [109]. In addition, for the renal injuries, MCP inhibited Gal-3 at the tubular level but not in the glomeruli, which is highlighted by the authors since Gal-3 is not expressed at the glomerular level [109]. Nephrotoxicity is a major side-effect of cisplatin chemotherapy, which contributes to the number of acute kidney injury (AKI) hospitalizations. Mice that received 1% MCP in drinking water during the same period after cisplatin injection had better morphometric preservation and prognostic regarding renal fibrosis, such as lower serum creatinine levels, Gal-3, fibronectin, and collagen-1 expression, which could point to a protective effect of MCP in this treatment [110]. Specific mechanisms and routes of Gal-3 downregulation, however, were not successfully addressed by those studies. Therefore, those points should be considered for better knowledge of the possible link between treatment and the biological effect.

AKI-induced mice through the ischemia/reperfusion (IR) model led to Gal-3 expression, cardiac injury, and systemic inflammation. The authors indicate that the induction of inflammation alongside cardiac fibrosis was Gal-3 dependent, demonstrated through significant reduced deleterious outcomes in genetically Gal-3-KO mice that received orally MCP (100 mg/kg/day), which was also seen at WT-MCP-treated mice group [111], but the expression of other markers such as MCP-1 and ICAM-1 mRNA were also decreased by MCP treatment. Rats submitted to myocardial IR also had better prognostics when treated with MCP (in drinking water) one day before and eight days after the procedure, such as improved perfusion, serum brain natriuretic peptide normalization, IL-1b and C-reactive protein (CRP) reduction, lower Gal-3 expression levels at the ischemic tissue, and other parameters. The Gal-3 specific blockade suggested was measured by the authors through the expression of two proteins that are down-regulated by Gal-3, fumarase, and reticulocalbin-3, which were restored after MCP treatment [112]. Perindopril and MCP (in drinking water) were similarly effective as treatments for ischemic heart failures in rabbit models, lowering Collagen-I, III, and Gal-3 mRNA and protein expression, alongside slight reversion of histological remodeling (an important level of fibrosis was still maintained, visible by Masson staining of myocardial tissues). However, the exact mechanisms by which MCP or perindopril could exert their observed effects are still unclear and were not directly addressed [113]. Both the effects of Gal-3 and isoproterenol-induced left ventricular systolic dysfunction in the mice model with selective hyperaldosteronism, alongside myocardial fibrosis installation. Combined therapies with MCP in drinking water and canrenoate as an aldosterone blocker enhanced the anti-inflammatory and anti-fibrotic effects [114]. Rats in a pressure overload model that were treated with MCP in drinking water had lower Gal-3 mRNA expression and Gal-3 immunostaining-confirmed presence than control. Other fibrosis-related proteins such as α-smooth muscle actin (α-SMA), connective tissue growth factor (CTGF), transforming growth factor (TGF)β-1 and fibronectin, and also inflammation factors such as IL-6, IL-1β, and TNF-α were at lower levels than control [115]. The studies described herein regarding oral MCP supplementation, either isolated or in combination with other molecules: (i) did not only show effects influencing Gal-3, but also other pathways, receptors, and protein expressions; (ii) authors did not explain, or at least brought up for the discussion, if orally consumed MCP could reach the target organs, such as kidney, heart or liver. These factors should be considered whenever reading these interesting but overly biased results, denoting not-so exclusive observations that are also promising but opening many questions regarding the systemic distribution of pectin’s pre- or post- colonic fermentation.

As mentioned before, there are physiological activation pathways that could involve Gal-3 action, important data to account for whenever testing new possible therapeutic agents that could selectively induce activation/apoptosis depending on the target [116]. This in vitro work has demonstrated that recombinant human Gal-3 initiates three distinct pathways, one for T cell activation (PIK3) and two hybrids (reactive oxygen species and protein kinase C—ROS and PKC, respectively). Furthermore, MCP and acidic fraction from ginseng roots inhibited T cell apoptosis in vitro by caspase-3 cleavage, and ginseng-derived fractions did not interfere in IL-2 secretion [116]. One of the Gal-3 connections established by the authors to the observed effects is proposed as upregulation of PI3K/Akt phosphorylation, in which the presence of an inhibitor for each molecule also inhibited IL-2 secretion. However, no similar effects regarding the effects of MCP in cardiac protection, fibrosis regulation, and normalized hypertension were observed in a recently published clinical trial from Lau et al. [117] in patients with high Gal-3 levels and established hypertension. The results raise many doubts and counterpoints regarding the MCP effects in in vitro and in vivo studies to be replicated in humans, which is discussed in the next chapter.

Recently, Gal-3 has been treated as a treatment target and prognostic marker for patients with severe acute respiratory syndrome-coronavirus 2 (SARS-CoV-2). Higher serum levels of Gal-3 were related to a tendency of severe acute respiratory distress syndrome (ARDS) development, and alongside IL-6 and CRP, Gal-3 was demonstrated to be the best predictive power for mortality outcome [118]. There are also positive correlations between Gal-3 and other inflammatory markers such as PTX-3, ferritin, and the marker of endothelial dysfunction, sFlt-1 [119], which could be utilized for intensive care unit (ICU) admission biomarker. A phase 2a study, the first clinical trial using an inhalator treatment targeting Gal-3-GB0139—associated with the standard of care procedure (dexamethasone)—identified lower Gal-3 serum levels, higher mean downward of CRP levels (although higher CRP was identified at first in the treatment group) and other inflammatory agents and better fibrosis marker levels than the control group. However, there was no statistical difference between patient mortality rates between groups [120]. This enhances and augments the discussion to further explore alternative treatments regarding Gal-3 inhibition.

## 4. Pectin and Gal-3 Controversies

As it has been widely studied and known, pectins are extensively fermented through the local intestinal microbiota, which could systematically impair “direct” mechanisms of action. In the above-discussed works showing systemic effects after oral ingestion of MCP, it is often ignored the thought process and viability on how those molecules could reach the systemic circulation. In counterpoint, a work that used antibody recognition of RG-I fragments from *Bupleurum falcatum* L.—demonstrating reactivity at mice bloodstream and liver—could indicate a partial small-intestine related absorption [2,121]. Similar studies based on β-glucan uptake mechanisms should be more explored for pectins in general, as both are non-digestible carbohydrates that suffer fermentation to an extensive degree [2]. Modified pectin from broccoli (*Brassica oleracea* L. Italica) was suggested to be absorbed after an increased number of activated macrophages and lymphocyte proliferation when administered through oral treatment in mice, but with no same effect in vitro [122]. One possible suggestion of pectin “absorption” is related to asialoglycoprotein receptors that could play a role in absorbing modified pectin fragments throughout the intestine, as they are notable galactoside-terminal glycoproteins transporters [123,124]; however, much more pieces of evidence need to be described to confirm this hypothesis. The oral consumption of MCP reduced liver metastasis on a mouse colon cancer model [125], while MCP-derived galactans ([-4-β-d-Gal*p*-1-]n) and arabinans ([-5-α-L-Ara*f*-1-]n) with a low degree of polymerization were absorbed through paracellular transport, and in lower rates through transcellular transport, similar to what was observed at in vitro culture of Caco-2 monolayer models [126]. Pectin-derived oligosaccharides with 1 kDa and rich in galactosyl residues were absorbed in BALB/c mice and human tumor cells while also changing membrane permeability in different human cancer cells, such as HepG2 and Colo 205 (hepatic and colon carcinoma, respectively) [127]. Microfold cells and gut-associated lymphoid tissue (GALT) are also proposed as explanations for the bloodstream presence of modified pectin fragments, where the former theoretically would serve as a facilitator to GALT macrophages to act internalizing these portions [122,128]. Once again, these assumptions of absorption models are still in germinative steps; therefore, they cannot be taken as unreasoning facts, but they can open new potential transcription elucidation of in vivo observed effects towards clinical significance in the future.

The above described highly selective effects, although not completely mechanistically understood, could be related to the mentioned diverse binding sites and different chemical conformations between those polysaccharides and Gal-3, but the range of varied mechanisms unrelated to Gal-3 specific inhibition cannot be excluded or ignored, as it could also play a role or even be the major protagonists regarding polysaccharides action.

In a related but slightly different context, it is highly important to clarify that a binding molecule is not the same as an inhibitor by itself. Additionally, this capability of pectins to exert both instances, binding to or inhibiting Gal-3, has been contested. As demonstrated by Stegmayr et al. [50], the use of a range of plant polysaccharides to study the capacity of interacting with representatives of the galectin family (including Gal-3) resulted in a very low or even absent binding and agglutination inhibition [50]. It is further discussed that the discrepancy observed in similar studies, such as the one from Gao et al. [95], could be due to a fine-tuning difference (concentration and temperature) or even multivalence interaction factors [50,95]. Furthermore, noteworthy, immobilized surface techniques may be prone to developing suitable conditions to the multivalent aspect, therefore potentially overestimating affinity results [88]. Nevertheless, the same authors also found a potential indirect effect. Re-incubating the JIMT-1 cells with the pectin samples led to Gal-3 accumulation around intracellular vesicles, feasibly inclining towards a “directional” change of location inside the cell, although more experiments regarding this property need to be performed to detect plausible applications of this observed scenario in vivo [50]. Other studies demonstrated biological effects in different in vitro cell cultures were totally or highly independent of Gal-3 inhibition; as demonstrated through Gal-3 hemagglutination (G3H) assay, cell lines with low expression of Gal-3 and/or usage of lactose (Gal-3 inhibitor) did not influence HG or RG-I activities, suggesting that cell migration inhibited by those polysaccharides did not rely upon human/mouse Gal-3 binding [8,129]. The main problem observed in literature is the overused statement of plant polysaccharides acting as specific pharmacological direct inhibitors of Gal-3, without specific data of inhibition shown, or even proposed study models to evaluate the integrity of those polysaccharides in reaching potential target organs. The facts were described in the letters written by Hakon Leffler, MD, Ph.D., and Anwen Shao, MD, Ph.D. [130,131], in response to pectin attribution of Gal-3 inhibition at blood–brain barrier disruption, relying on the lower expression detected by immunoblotting [132], but this is only one of the examples. Specific human Gal-3 inhibition of the S-face CRD region, mostly between C and D subsites, is achieved mainly by small glycoconjugates, such as lactose, N-acetyllactosamine, and many synthetized neo-glycoproteins [30,49,133,134,135,136].

To address those controversial perspectives, more studies regarding alternative mechanisms independent of Gal-3 interaction should be performed with polysaccharides while also exploring more structure-relation models. In the following chapters, alternative perspectives are further analyzed.

## 5. Pectin as Dietary Fiber: Some of the Gal-3 Independent Beneficial Effects to Human Health

Regarding pectin fermentation, it generates products that are essential for colonic and systemic health in general. One example is the inhibition of cholesterol intestinal absorption in an apoE^−/−^ mice model through regulating mRNA levels of its transporters, resulting in controlled blood lipid levels in vivo [10]. This protection is also due to the physicochemical properties of the soluble fibers, where the bile acid excretion and cholesterol mobilization in the intestinal tract is compromised by the fiber viscosity [137]. Similar results were obtained in mildly hypercholesteremic humans, in which pectin with high DE and high molecular weight resulted in a cholesterol-lowering effect [12]. A rodent model of high-fat diets to induce non-alcoholic fatty liver disease also had positive results after receiving 8% citrus peel pectin. The diet had attenuated liver damage and lipid accumulation while also reducing some biomarkers such as carbohydrate-responsive element-binding protein (ChREBP) and reducing serum total triglyceride in vivo [11].

Regarding glucose metabolism, pectin (from apple and citrus) added to high fat/high sugar diets, even at low doses, were successful at ameliorating glucose serum levels [13,14], glucose tolerance, and insulin resistance biomarkers such as HOMA-IR and fasting insulin serum levels [13]. Those effects are suggested to be derived from a pectin capacity of lowering mucosal disaccharidase activities, specifically sucrase and maltase [14], and also through a potential pectin p-AKT upregulation, being beneficial to directing the insulin signaling [13].

Short-chain fatty acids (SCFA) are the most common by-product of pectin and other types of fibers fermentation, and there is extensive literature related to biological and health effects [138,139,140,141]. The SCFA help to maintain intestinal health through G-protein-coupled receptors (GPR) interaction, such as T regulatory cell homeostasis, epithelium integrity, and maintenance of an acute immune response and normal cytokine/chemokine expression of key modulators [142,143,144,145]. Different types of pectins and fragments can modulate microbiota and have different fermentation profiles. Sugar beet and soy pectin lowered *Akkermansia* relative abundance, while soy pectin showed high levels of propionate, butyrate, and branched SCFA concentrations in the colon of rats [146]. Supplementation of citrus pectin (CP) in piglets diets was also attributed to a higher relative abundance of Bacteroidetes members in colonic digesta and feces, and this pectin-enriched fraction also slowed the fermentation process, changing microbiota interaction [147]. In a dynamic digestion/fermentation simulator, CP could induce *Bifidobacterium* spp. growth, but not *Lactobacillus* spp. Both genera presence are considered beneficial to colonic health [148]. During in vitro fermentation, authors had similar results with high DM (70%) apple pectin [149], with in vivo observations also supporting the mentioned in vitro data [11]. The CP was capable of inducing growth of *Faecalibacterium prausnitzii*, a bacteria that has been pointed as a modulator in dysbiosis of Crohn’s disease patients and a major agent of pectin utilization, with their lower levels correlated to inflammatory bowel disease [150,151,152]. Those data support the application of pectins in health investigations, even before going deeper into specific binding/modulation features, and they are summarized in Table 2 (for the diverse discussed biological effects of pectin).

## 6. Should Gal-3 Inhibition Be the Main Biological Effect Expected from Pectin?

Although the potential Gal-3 inhibition achieved by pectin could be a resourceful knowledge area, there is a great extent of literature showing parallel ways. Wild-type and Gal-3 knockout (Gal-3^−/−^) HCT-116 (human) cells were exposed to different papaya pectin fractions in vitro. Specifically, the most acidic fraction (uronic fraction from fourth day after harvest), with a mean molecular weight of 128 kDa and high antibody reactivity to LM5 and LM16 (1,4-β-galactan and type-1-rhamnogalacturonan, respectively), kept a slightly lower efficacy at the Gal-3^−/−^ cells, suggesting that even though the cancer cell viability decrease could be in part due to Gal-3 inhibition, it was not the only molecular modulated pathway [7]. Commercially available CP and MCP demonstrated to have a pro-inflammatory action independent of Gal-3 inhibition, upregulating cytokine secretion in the spleen of BALB/c mice, IL-17, IFN-γ, and TNF-α through IL-4 [153]. Modified pectin obtained from *Theobroma cacao* pod husks, highly composed of uronic acids, galactose, and rhamnose, with a low degree of methylation and amidation, was also attributed with a pro-inflammatory profile, similar to LPS stimulation in isolated macrophages from mice, upregulating secretion of IL-12 and TNF-α, although stimulating the anti-inflammatory IL-10 simultaneously. The pectins, however, did not enhance the phagocytic activity of the peritoneal macrophages [154]. Similar effects were detected in differentiated macrophage (THP-1) cell cultures that were in contact with native sweet pepper pectin. The polysaccharide was characterized as pectin composed of uronic acids, galactose, and arabinose, also confirmed through NMR with signals of methyl and acetyl groups linked to α-D-Gal*p*A with a high degree of methylation (85%) and low degree of acetylation (5%). There were also signals of (1→4)-linked-β-d-Gal*p* units, which the authors attributed to type-1-arabinogalactans. The sweet pepper pectin induced in vitro TNF-α, IL-1β, and IL-10 secretion at the highest concentration used [155]. After modifying the native structure and removing its side chains by partial acid hydrolysis, the composition was uronic acids (91%) and rhamnose (9%). The respective signals identified in the native molecule related to galactan core residues had disappeared, with also a great reduction in DM (down to 17%). This modified sweet pepper pectin was still able to induce—at a lower rate than native pectin—the TNF-α and IL-10 secretion but at a higher rate the IL-1β [155]. Here, it is again highlighted the importance of the structure-dependence of pectic fractions with different potential targets. In earlier studies, sweet pepper pectin (1,4-α-d-galacturonan partially substituted with methyl and *O*-acetyl backbone) and low methoxyl CP performed similarly by lowering TNF-α and enhancing IL-10 secretion, which resulted in ameliorated survival rate in endotoxin-shock induced mice models [102,156]. Elsewhere, CP was also capable of reducing LPS-induced hypothermia and inflammatory cytokine gene expression, suppressing IL-6 production and TLR-4 signaling in vivo [157].

Toll-like receptors (TLR) have been thoroughly explored, especially the dichotomy of weighing between agonists or antagonists for cancer treatment, where the same TLR can exhibit anti or pro-tumor immune responses [164,165,166]. Lemon pectin ranging from 40 to 100 kDa and from 30 to 74% of DM, activated in vitro THP-1 phagocytic cells depending on *MyD88* in a TLR-mediated manner [158]. Additionally, the pectin DM and its structural backbone were correlated to NF-κB/AP-1 activation through TLR, where highly esterified polymers were strong activators, and their oligomers (produced after extensive hydrolytic processing) did not perform equally [158]. Similarly, ginseng polysaccharide extract composed of RG-II was a TLR-4 up-regulator and MyD88 activator in a structure-dependent manner in vivo [116]. A polysaccharide extracted from a European berry (*Hipphoides rhamnoides* L.) with 85% DM and consisting of repeating units of (1→4)-β-d-galactopyranosyluronic acid residues strongly stimulated TLR-4 [159,167]. The blockage of TLR4/MyD88 was used by authors to explore the interaction between the pectic structures and macrophages, where it inhibited the increase observed in nitric oxide and other cytokines induced by treatment with the polysaccharides [167]. Ginseng RG-II stimulated TLR-4, increasing dendritic cell maturation and activation through induced cytokine and mitogen-induced protein kinases (MAPKs) production. This cytotoxic T cell response inhibited tumor growth of EG7 lymphoma cells [160]. Interestingly, the ripening process for some sources of pectin, such as papaya, can be of utmost importance regarding main structural changes to effectively interact with diverse TLR receptors [73]. The authors demonstrated that although TLR-2 and -4 were activated by pectins isolated from unripe and ripe papayas, TLR-3, -5, and -9 were not activated by the pectins isolated from unripe fruits in two different time points (pectins with 580 and 610 kDa, higher galactose and glucose contents, alongside with a proportionally lower composition of GalA residues; the second unripe point had 15% DM). Specifically, TLR-3 and -9 could be inhibited by pectins isolated from unripe papayas due to high molecular weight structures [73], similar to what was previously observed with lemon pectins that inhibited the TLR-2 heterodimer formation with TLR-1, but not with TLR-6 [161]. Citrus pectin ranging from 18 to 69% DM (but without further structural details or suggestions) was able to inhibit TLR/MyD88 by oral administration in mice and decreased TLR-2 mediated immune response in rats when administered both in alginate microcapsules and directly at drinking water [162]. Administration of pectin samples and usage of checkpoint inhibitors could be powerful measures to assure immune responses in certain types of cancer [168]. Neohesperidin nanoliposomes incorporated with citrus pectin (65% DM) and chitosan (50 kDa, degree of deacetylation of 85%) had higher cellular uptake rates in comparison to chitosan or neohesperidin single treatments [169]. A nephropathy murine model with MCP added in drinking water found protective effects of the pectin sample independent of Gal-3 inhibition at early proliferation, but with Gal-3 downregulation later on [163]. An overall scheme of the possible biological effects of pectins that were discussed throughout the paper are depicted in Figure 6 using an intestine model for easier comprehension, and the different biological effects observed in in vitro, in vivo, and cohort/clinical studies are summarized as a table (Table 2).

Even though extensive human model studies are needed to confirm the positive outcomes of pectin ingestion regarding specific targets, the data obtained until now can help to demonstrate a variety of positive health-regulating effects of pectins. The pectin binding and inhibition of Gal-3 is one of them; however, this relationship is still highly controversial and contested. Human Gal-3 inhibition and the consequent beneficial effects on human health must be covered, from in vitro experiments to human clinical trials, passing through the bioavailability assays of pectin-derived fragments in the human body. The plausible wider range of usage for the polysaccharides is highlighted in this revision, enhancing the discussion of the inclusion of pectin molecules for synergistic effects with different molecules and drugs. Moreover, pectin could exert direct and indirect immunomodulatory effects, and their fragments could interact with different types of receptors. The fermentation products of pectins also help to sustain the intestinal and systemic environment, with all these possible beneficial effects of pectins being potentially achieved through supplementation by oral intake.

## 7. Conclusions

Targeted galectin-3 binding sites for therapeutic approaches are diverse. Pectin is an important food component classified as soluble dietary fiber. Its biological effects on human health go from colonic fermentation and microbiota modulation to potential direct interaction with intestinal cells and proteins, such as TLRs and Gal-3. Many studies can effectively suggest the multi-way interaction between pectin molecules and these ligands. The Gal-3 binding is suggested in the CRD motif, both through its F-face and S-face sites. Furthermore, the binding can occur in the N-terminal tail and even in a two-step interaction method, promoting subtle motif variations in the protein molecule and therefore enhancing an adequate interconnection with the pectin molecules. However, the biological effects of pectin transcend Gal-3 interaction and/or inhibition, which is far from being an established point, with several challenges to be overcome, and has undergone valid confrontations. All the literature and methodology improvements converge on the pectin diversity for enhancing human health. Moreover, pectin molecules exerting distinct regulatory/inhibitory effects, low side effects, and natural sources are interesting bioactive components to be added into dietary supplements. These pectin molecules could be used continuously to increase the natural intake of bioactive polysaccharides, e.g., in post-cardiac arrest and renal fibrosis pathologies, as well as to a great extent as an auxiliary factor in chemotherapy and possibly as immunomodulatory molecules. Extensive additional research is needed before confirming any of those promising illustrated scenarios.

## Figures and Tables

**Figure 1 biomolecules-12-00289-f001:**
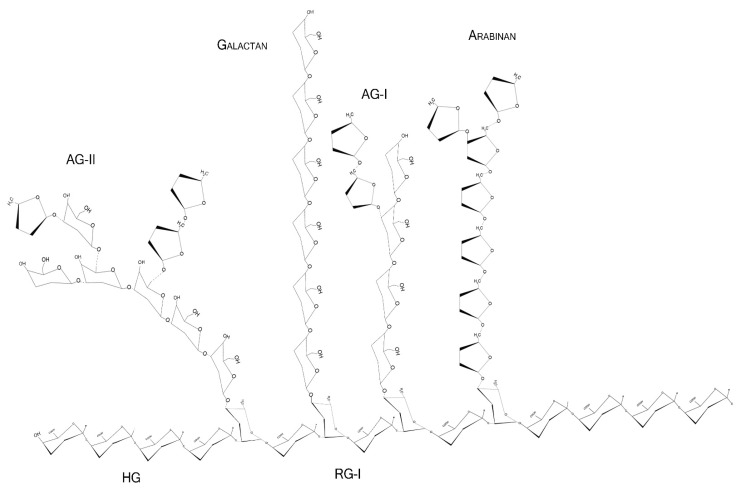
Schematic representation of major pectin components in chair conformation. HG—homogalacturonan, composed of linear α-1,4-d-galactopyranuronic acids; RG-I—intercalated α-d-galactopyranuronic acids and α-l-rhamnopyranose through α-1,4 and 1,2 glycosidic bindings; AG-I—β-1,4-d-galactopyranose with occasional *O*-3 α-l-arabinofuranose; AG-II—β-1,3-d-galactopyranose with *O*-6 α-l-arabinofuranose/arabinogalactans. Arabinans and galactans consist of linear α-1,5-l-arabinofuranoses and β-1,4-d-galactopyranoses, respectively.

**Figure 2 biomolecules-12-00289-f002:**
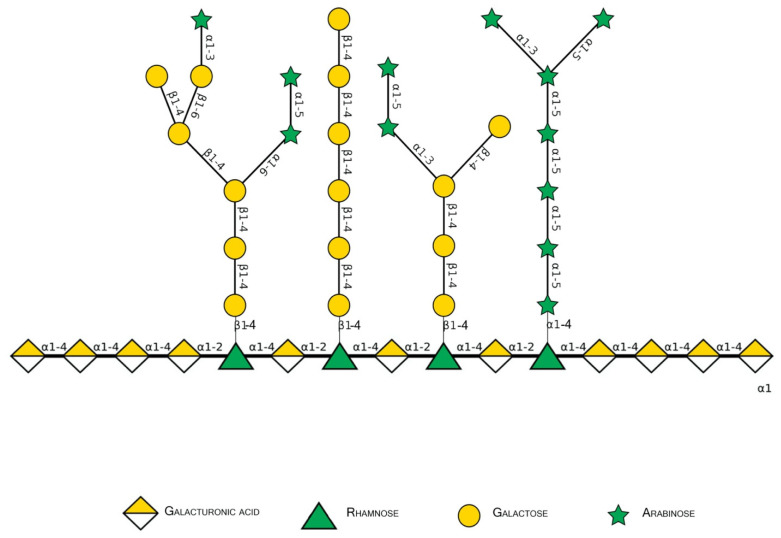
Schematic representation of major pectin components in Symbol Nomenclature for Glycans (SNFG-B) [9] model. Different glycosidic bounds common to pectins are illustrated.

**Figure 3 biomolecules-12-00289-f003:**
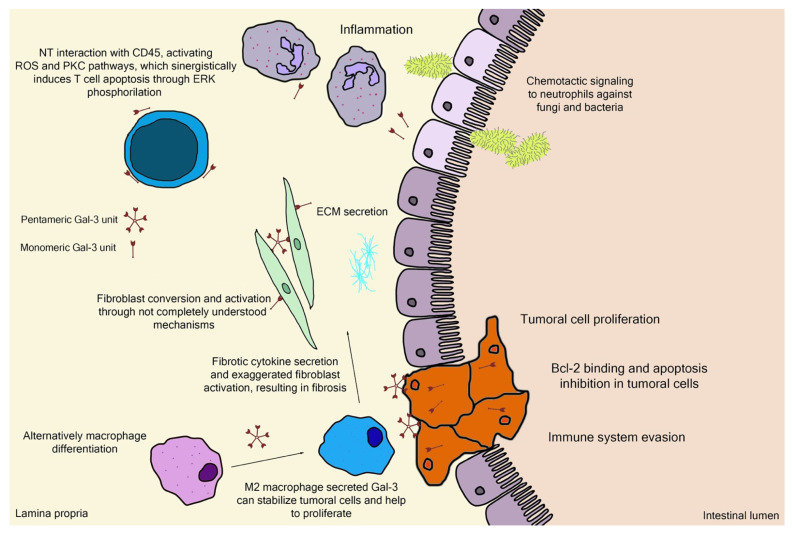
Schematic representation of (patho)physiological effects of galectin-3. As it is demonstrated, the physiology is separated by a thin line from the pathological scenario, such as the extracellular matrix (ECM) secretion stimuli or the chemotactic signaling for infiltrative immune cells. Bcl-2—B-cell lymphoma 2; ERK—extracellular signal-regulated kinases; NT—N-terminal tail/domain; PKC—protein kinase-C; ROS—reactive oxygen species.

**Figure 4 biomolecules-12-00289-f004:**
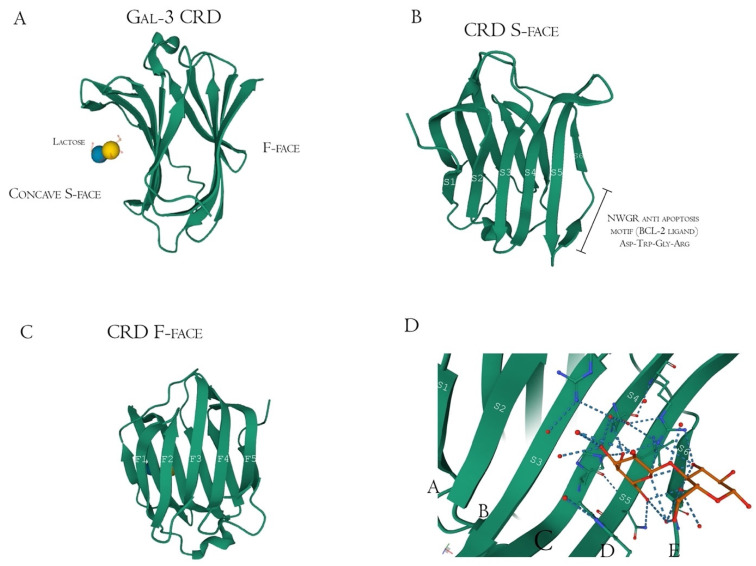
Galectin-3 crystalized tertiary structure X-ray diffraction, PDB ID 49RB [75,80,81]. (**A**) Gal-3 complete CRD, with the anti-parallel β-sheet sandwich; (**B**) CRD S-face β-sheets (S1–S6), which holds the ABCDE subsites. The NWGR motif is highlighted because of their biochemical importance; (**C**) CRD F-face β-sheets (F1–F5, also numbered as B-9, -8, -7, -2, and -11, respectively); (**D**) Schematic representation of β-d-galactopyranosyl-1,4- β-D glucopyranose (β-Lactose) binding at the canonical S-face. The hydrogens atoms are colored as red, the hydrogen bonds as blue dotted lines, and the lactose chain as orange. The binding is stronger at the C subsite (between S4 and S5) amino acids and the β-d-galactopyranosyl residue.

**Figure 5 biomolecules-12-00289-f005:**
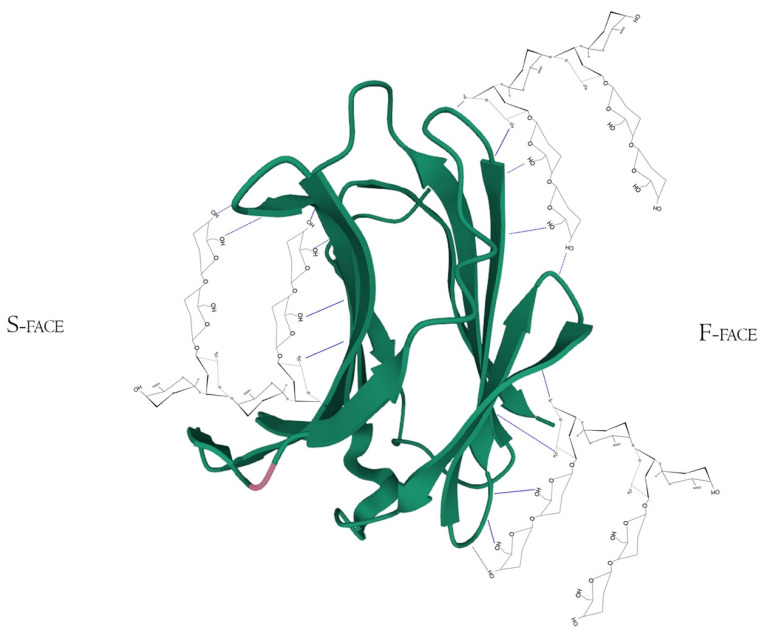
Hypothetical interaction of RG-I fragments with both F and S-face of Galectin-3 simultaneously. Here, the main protagonists would be the β-d-galactopyranose and α-d-galactopyranuronic acid residues and would not act like specific pharmacological inhibitors, but maybe as potential Gal-3-ligand blockers through multivalency or allosteric occupation. Hydrogen bonds are represented by the blue dotted lines. PDB ID 49RB [75,80,81].

**Figure 6 biomolecules-12-00289-f006:**
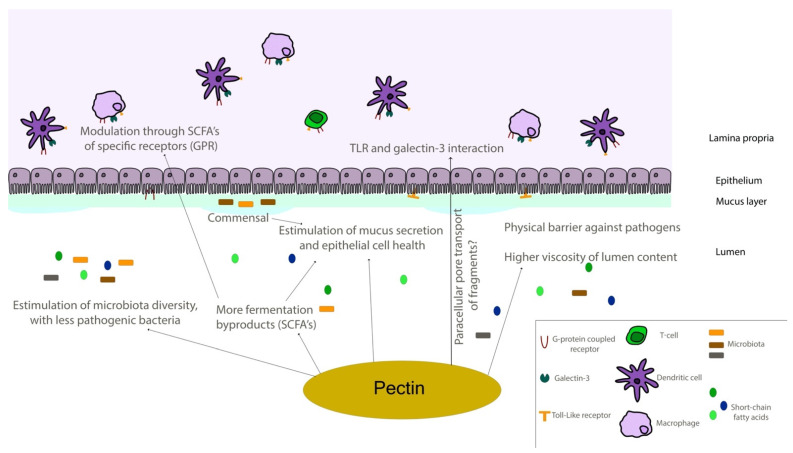
Schematic representation of the intestinal environment. Pectin molecules can interact in different ways with epithelial and immune components of intestinal tissues, regulating different responses directly and through fermentation by-products. SCFA—short-chain fatty acid; TLR—toll-like receptor.

**Table 1 biomolecules-12-00289-t001:** Polysaccharide-Gal-3 binding articles summary.

Authors	Polysaccharide Residue	Analysis Method	Binding Evaluation
Wu et al., 2020 [33]	RG-I from citrus canning process water	Surface plasmon resonance	Smooth binding curve through SPR with decreased affinity with galactan side-chain removal
Zhang et al., 2016 [34]	MCP, RG-I-4, and p-galactan	Gal-3 hemagglutination, bio-layer interferometry, and surface plasmon resonance	RG-I-4 demonstrated higher Gal-3 avidity in comparison to the other two polysaccharides, with a K_D_ at sub-micromolar range (RG-I-4 and p-galactan), but no significant result when testing competitive assays with known S-face inhibitors such as lactose
Gao et al., 2013 [40]	Ginseng RG-I-4 domain	Gal-3 hemagglutination and surface plasmon resonance	RG-I-4 inhibited G3H and was bound specifically to CRD with high affinity with Ara residue location in the RG-I, changing the activity detected at the G3H assay
Gunning, Bongaerts, Morris et al., 2009 [51]	RG-I, PG, and galactans	Atomic force microscopy, fluorescence microscopy, nuclear magnetic resonance, and flow cytometry	Galactan binding to Gal-3 is lectin-saccharide highly specific, while RG-I has low specificity, and PG was not specific. The data suggest that the lesser “sterical crowding” of the galactans alongside its beta-1,4 linear chain could be the reason for the better performance observed
Shi et al., 2017 [52]	Ginseng RG-I-3A domain	Bio-layer interferometry, Gal-3 hemagglutination	Binding kinetics of RG-I-3A showed a high binding affinity with a K_D_ of 28 nM through and also presented notable G3H inhibition
Zhang et al., 2017 [83]	MCP-derived RG-I and HG portions	Gal-3 hemagglutination, bio-layer interferometry, ELISA, and nuclear magnetic resonance	Gal-3 bound to both portions separately but with a much more notable avidity when a combination of them (RG + HG) is performed, suggesting that this interaction exposes more binding sites at the lectin
Miller et al., 2015 [84]	Galactomannans (GM) and polymannan	Nuclear magnetic resonance	The primary binding surface of the GM’s located mainly at F-face beta-sheets (7,8 and 9)
Zheng et al., 2020 [89]	MCP-derived HGs of varying molecular weights	Nuclear magnetic resonance heteronuclear single quantum coherence spectroscopy and crystallography	Higher molecular weight HGs demonstrated more perturbances at F-face resonances and involved more S-face beta-sheets at the binding footprint. A possible binding of Gal-3 to the non-terminal HG sites is suggested, and it is shown a different S-face binding pattern of HG’s compared to lactose
Miller et al., 2019 [90]	Galactan oligosaccharides of varying chain lengths	Nuclear magnetic resonance heteronuclear single quantum coherence spectroscopy	Binding affinity at the terminal non-reducing end of the galactans in the CRD S-face (beta-sheets 4, 5, and 6 chemical shifts mostly) increases with the increase in chain length
Zhao et al., 2017 [91]	Pumpkin RG-I-containing pectin	Surface plasmon resonance	Moderate binding affinity towards Gal-3 through SPR, with a fast association between protein and polysaccharide (K_A_) and slow dissociation (K_D_)
Miller et al., 2017 [92]	Ginseng RG-I-4 domain	Nuclear magnetic resonance heteronuclear single quantum coherence spectroscopy	Epitopes from RG-I-4 bind to three different labeled Gal-3 sites, two at the CRD and another one at NT. At lower concentrations, the F-face site is more activated, turning to S-face at higher ones

**Table 2 biomolecules-12-00289-t002:** Summary of observed experimental effects in manuscripts studying pectin and its fragments.

Authors	Treatment	Study Type	Treatment Target	Observed Experimental Effects
Pedrosa, Lopes and Fabi, 2020 [7]	Papaya pectin acid and neutral fractions	In vitro	HCT 116, HT-29, and HCT-116 Gal-3^−/−^	Gal-3-mediated agglutination inhibition, cell viability decrease in both WT and knockout cells (suggesting Gal-3 independent pathways)
Chen et al., 2018 [10]	SCFAs	In vivo	Male apoE^−/−^ mice	Stimulation of Lxrα mediated genes expression related to intestinal cholesterol uptake and excretion; improved blood lipid profiles and anti-atherosclerotic property
Li, Zhang, and Yang 2018 [11]	CP	In vivo	Healthy male C57BL/6J mice	Pectin-supplemented high-fat diet mice had reduced lower liver damage, lipid accumulation, and total serum triglyceride
Brouns et al., 2012 [12]	Different DM and MW apple and citrus pectin (CP)	Human intervention	Mildly hyper-cholesterolemic men and women	Higher DM apple and citrus pectin lowered between 7 and 10% low-density lipoprotein cholesterol (LDL-C) compared to control
Liu et al., 2016 [13]	CP	In vivo	Male Sprague-Dawley rats with induced type 2 diabetes	Enhanced glucose tolerance, blood lipid levels, reduced insulin resistance, pAKT expression upregulation, and glycogen synthase kinase 3 β (GSK3β) downregulation
Fotschki et al., 2014 [14]	Apple fiber (low pectin)	In vivo	Male Wistar rats	Disaccharidase activity reduction, higher SCFA production, reduced serum glucose concentration
Prado et al., 2019 [32]	Chelate-soluble fraction of papaya pectin	In vitro	HCT 116 and HT-29 human colon cancer cells	Gal-3-mediated agglutination inhibition, similar to lactose control; pre-treatment with lactose suggests cell Gal-3 independent proliferation reduction for one of the fractions (3CSF)
Wu et al., 2020 [33]	CP fragments	In vitro	MCF-7 human breast cancer and A549 human lung carcinoma	Significant binding affinities to Gal-3; dose-responsive cell proliferation inhibition in vitro, not necessarily related to Gal-3
Gao et al., 2013 [40]	MCP, ginseng pectin fractions, potato galactans, and RG-I	In vitro	HT-29 human colon cancer cell line	RG I-4 from ginseng strongly inhibited Gal-3 mediated hemagglutination; better inhibition of cell adhesion and homotypic cell aggregation than lactose
Stegmayr et al., 2016 [50]	MCP	In vitro	JIMT-1 breast cancer cells	No Gal-3 inhibition was detected; however, MCP pre-incubation resulted in the accumulation of Gal-3 molecules around intracellular vesicles
Prado et al., 2020 [73]	Papaya pectins from different ripening periods	In vitro	THP-1 human monocytic cell	Different TLR’s activation and inhibition depend on the ripening period
Hu et al., 2020 [85]	Lemon pectin	In vitro	Human pancreatic beta-cell	Unspecific and unspecified reduction of deleterious effects of inflammatory cytokines with very low (5%) degree of esterification pectin at cell culture
Xu et al., 2020 [86]	MCP	In vivo	Male Wistar rats	Down-regulation of Gal-3, TLR, and MyD88, decreased expression of IL-1β, IL-18, and TNF-α
Maxwell et al., 2016 [99]	Sugar beet and CP	In vitro	HT-29 human colon cancer cell line	Cell proliferation control and induction of apoptosis
Pynam and Dharmesh, 2019 [101]	Bael fruit pectin fragments	In vitro and in vivo	Healthy Swiss albino mice and B16F10 cell line	Microbiota protection, tyrosinase down-regulation, Gal-3 binding, downregulation of Gal-3 gene, IL10 and IL17 cytokines
Fang et al., 2018 [103]	MCP	In vitro	Human urinary bladder cancer (UBC) cells	Gal-3 down-regulation and inactivation of Akt signaling pathway, a decrease in Cyclin B1, G2/M phase arrest, Caspase-3 activation
Hossein et al., 2019 [104]	MCP	In vitro	SKOV-3 and SOC (serous ovarian cancer) cells	Synergistic effect of PTX and MCP increasing caspase-3 activity and decreasing cyclin D1 expression level
Abu-Elsaad and Elkashef, 2016 [105]	MCP	In vivo	Adult male Sprague-Dawley rats	Decreased liver fibrosis and necroinflammation, a decrease in MDA, TIMP-1, Col1A1, and Gal-3, increase in Caspase-3, gluthatione, and superoxide dismutase expression
Martinez-Martinez et al., 2016 [106]	MCP	In vivo	Adult male Wistar rats	Attenuation of renal fibrosis-related biomarkers, osteopontin, cytokine A2, albuminuria and TGF-β1
Calvier et al., 2015 [109]	MCP	In vivo	Adult male Wistar rats, C57BJ6 WT and Gal-3^−^^/^^−^ mice	Reverted fibrosing markers and Gal-3 augmentation levels, similarly to spironolactone
Li et al., 2018 [110]	MCP	In vitro and in vivo	HEK293 cells and C57BL/6 male mice	Amelioration of renal interstitial fibrosis, lower collagen I and fibronectin in the kidney, reduced IL-1β mRNA levels, lower Gal-3 expression
Prud’homme et al., 2019 [111]	MCP	Cohort and in vivo	C57BL6/J and C57BL6/J Gal-3 KO male mice	Cardiac fibrosis induced by model prevented by MCP treatment, IL-1β level maintained, protected, treated mice against renal inflammation
Ibarrola et al., 2019 [112]	MCP	In vivo	Male Wistar rats	BNP serum level normalization, lower Gal-3 cardiac expression, reticulocalbin-3 and fumarase in the myocardium, IL-1β and CRP in serum
Li et al., 2019 [113]	MCP and perindopril	In vivo	New Zealand male rabbits	Gal-3, collagen I, and III downregulation
Vergaro et al., 2016 [114]	MCP	In vivo	Transgenic mice with aldosterone synthase gene overexpression	Reduced cardiac hypertrophy, fibrosis, Coll-1, and Coll-3 genes expression and also enhanced anti-inflammatory and anti-fibrotic effects when synergistically acting with Canrenoate
Ibarrola et al., 2017 [115]	MCP	In vivo	Male Wistar rats	Gal-3, mRNA expression normalized, collagen I, fibronectin, α-SMA, TGF-β1, and CTGF mRNA expression reduced compared to pressure overload group, vascular inflammatory markers expression was also controlled
Xue et al., 2019 [116]	Ginseng pectin fractions	In vitro and In vivo	Jurkat (human leukemia cells) and male IRC mice	MCP inhibited IL-2 expression, and the three pectin fractions utilized reversed cleaved caspase-3 formation alongside lactose. MCP and ginseng pectins inhibited ROS production in vitro. Reduced tumor weight and increased IL-2 secretion in vivo
Lau et al., 2021 [117]	MCP	Interventional trial	Participants with high Gal-3 levels and hypertension	MCP had no impact regarding attenuating of cardiac-related risk factors
Busato et al., 2020 [122]	Broccoli stalks pectin	In vitro and in vivo	Female albino swiss mice and peritoneal exsudate cells	Macrophage activation and higher phagocytic activity; IL-10 presence was higher at peritoneal fluid in vivo, but not at in vitro model
Liu et al., 2008 [125]	MCP	In vitro and in vivo	CT-26 cells and Balb/c female mice	MCP did not alter Gal-3 expression at metastatic liver cells, although it did inhibit tumor growth and metastatic rate
Courts, 2013 [126]	MCP	In vitro	Caco-2 monolayer	MCP fragments were absorbed through paracellular and to a lower degree by transcellular transports at in vitro culture
Huang et al., 2012 [127]	Enzyme-treated CP	In vitro and In vivo	HepG2, A549, Colo 205, and HEK293 cells, BALB/c mice	Altered membrane permeability (LDH release) in the cancer cell lines; low weight oligogalacturonide was absorbed by the mice and the tumor cells, enhancing Gal-3 release to the medium
Fan et al., 2018 [129]	Ginseng RG-I enriched pectins	In vitro	L-929 fibroblast cells	Modulation of cell migration and adhesion, independent of Gal-3
Nishikawa et al., 2018 [130]	Modified citrus pectin (MCP)	In vivo	Male C57BL/6 mice	Attenuated blood-brain barrier disruption Gal-3 upregulation, inactivation of ERK 1/2, STAT and MMP
Sivaprakasam et al., 2016 [143]	2% inulin, 2% pectin, and 1% cellulose	In vivo	Human colon cancer tissue and Ffar-2^−^^/^^−^ C57BL/6J mice	Microbiota modulation, promotion of *Bifidobacterium* growth, and reduction of *Prevotellaceae*
Kim et al., 2013 [144]	SCFAs	In vivo	WT, GPR41^−^^/^^−^ and GPR43^−^^/^^−^ mice	Activation of intestinal epithelial cells to produce chemokines and cytokines, GPR’s were essential in T effector cell activation and signaling pathways
Tian et al., 2016 [146]	Sugar beet, soy, low DM, and high DM citrus pectin	In vivo	Male Wistar rats	More stimulation of *Lactobacillus* and *Lachnospiraceae* growth in sugar beet pectin, higher production of SCFA’s for low DM citrus and soy pectin
Tian et al., 2017 [147]	Low DM and high DM citrus pectin	In vivo	Piglets	The slower fermentation process, alteration of main fermentation region, and higher Bacteroidetes predominance
Ferreira-Lazarte et al., 2019 [148]	CP	In vitro	Dynamic gastric simulator with healthy volunteer fecal slurry donated	Growth stimulation of *Bifidobacterium* *spp*., *Bacteroides* *spp.*, and *Faecalobacterium prausnitzii*, high increase in acetate and butyrate production
Chen et al., 2013 [149]	Apple pectin oligosaccharides	In vitro	Fecal batch culture fermentation	Increased numbers of *Lactobacillus* and *Bifidobacteria*, a higher concentration of acetic, lactic, and propionic acid decreased number of Clostridia and Bacteroides
Onumpai et al., 2011 [150]	Potato galactan, methylated citrus pectin, beet arabinan, *Arabidopsis thaliana* RG-I	In vitro	Fecal batch culture fermentation	Higher *Bifidobacterium* populations and higher SCFA’s yield increased *Bacteroides-Prevotella* groups
Merheb, Abdel-Massih, and Karam, 2019 [153]	CP and MCP	In vivo	Female BALB/c mice	Upregulation of IL-17, IFN-γ, and TNF-α through IL-4 cytokine secretion in the spleen
Amorim et al., 2016 [154]	*Theobroma cacao* pod husk modified pectin	In vivo	Female albino Swiss mice	Promotion of macrophage differentiation, nitric oxide production, and upregulation of IL-12, TNF-α, and IL-10 secretion
Do Nascimento et al., 2017 [155]	Sweet pepper pectin	In vitro	THP-1 human monocytic cell	Modulation of TNF-α, IL-1β, and IL-10 production and secretion
Popov et al., 2011 [156]	Sweet pepper pectin	In vivo	Male BALB/c mice	Higher IL-10 production with lower TNF-α release
Ishisono et al., 2017 [157]	CP	In vivo	Male C57BL/6 mice	Suppression of IL-6 secretion from TLR activated macrophages and CD11c^+^ cells
Vogt et al., 2016 [158]	Different DM lemon pectin	In vitro	T84 intestinal epithelial cells	NF-kB/AP-1 activation through TLR/MyD88 and protective effects in the intestinal barrier
Wang et al., 2018 [159]	*Hippophae rhamnoides* L. berries pectin	In vivo	Cyclophosphamide induced immunosuppressive mice	Macrophage activation, MyD88 increased expression and upregulated expression of TLR4
Park et al., 2013 [160]	RG-II from P. ginseng	In vivo and In vitro	C57BL6 WT, TCR KO, TLR KO mice, and BMDC cells	Facilitation of CD8^+^ T cells, induced production of TNF-α, IL-12, IFN-γ, and IL-1β during dendritic cell maturation
Sahasrabudhe et al., 2018 [161]	Lemon pectins with different DM	In vitro and In vivo	HEK-Blue WT and mutated cell lines, female C57BL/6 mice	Inhibition of TLR2-1 heterodimer, prevention of ileitis in the mice model
Hu et al., 2021 [162]	Lemon pectins with different DM	In vivo	Sprague-Dawley male rats and C57BL/6 mice	Reduced peri-capsular fibrosis in vivo and decreased DAMP-induced TLR2 immune activation in vitro
Kolatsi-Jannou et al., 2011 [163]	MCP	In vivo	Male C57BL/6J mice	Reduced Gal-3 expression, reduced renal cell proliferation, apoptosis, fibrosis, and proinflammatory cytokine expression

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
