# Peer review of "The Complex Biological Effects of Pectin: Galectin-3 Targeting as Potential Human Health Improvement?"

_biomolecules, 2022, doi:10.3390/biom12020289_

Round 1

Reviewer 1 Report

In this review, the authors introduced the basic molecular characteristics of pectin in detail, including pectin molecular weight, monosaccharide composition, backbone and side chains, esterification degree, rheological properties and food source, and then introduced the Gal-3 binding site and the interaction with pectin, pectin and Gal-3 controversies, pectin as dietary fiber and other biological activities of pectin. This review systematically summarized the references about pectin and Gal-3, which is useful for researchers in the field.

The manuscript could be improved by revising some expressions and spelling, such as “sidechains”should be “side chains”. In Line 454, “Pectin’s are classified as-----” should be “Pectins are classified as ----“.   

Author Response

RESPONSE TO REVIEWERS

The authors thank the reviewers and the editor for their precise and valuable suggestions. We have modified the manuscript accordingly and have responded to all criticisms raised. We hope that now you will find it suitable for publication.

Reviewers' comments:

REV#1: In this review, the authors introduced the basic molecular characteristics of pectin in detail, including pectin molecular weight, monosaccharide composition, backbone and side chains, esterification degree, rheological properties, and food source, and then introduced the Gal-3 binding site and the interaction with pectin, pectin and Gal-3 controversies, pectin as dietary fiber and other biological activities of pectin. This review systematically summarized the references about pectin and Gal-3, which is useful for researchers in the field.

The manuscript could be improved by revising some expressions and spelling, such as “sidechains” “should be “side chains”. In Line 454, “Pectin’s are classified as-----” should be “Pectins are classified as ----“.

Answer: We thanks the reviewer for the suggestions. The manuscript was revised to improve expressions and spelling, such as those pointed by the reviewer.

Reviewer 2 Report

This review by de Freitas et al highlights the role of Galectin-3 inhibition in human health. The review is well-written, and reports interesting information through a comprehensive style.

I do not have major comments; they are mainly detailed oriented to help the authors to improve the paper.

  • I would suggest changing the title to facilitate the comprehension for the readers.
  • It could be interesting to reorganize some information.

For example, for MCP.

  • RENAL: MCP, by blocking Gal-3, is able to exert beneficial effects against renal fibrosis also in cisplatin-induced nephrotoxicity (doi: 10.1042/BSR20181803), hypertension  or acute kidney injury (doi: 10.1016/j.jacbts.2019.06.005).
  • MCP improved cardiovascular alterations in ischemia-reperfusion (doi: 10.1038/s41598-019-46119-6), ischemic heart failure (DOI: 10.3389/fphys.2019.00267), obesity or pressure overload(DOI: 10.3390/ijms18081664). Moreover, the effect of Galectin-3 inhibition with MCP in hypertension has been recently published (doi: 10.1016/j.jacbts.2020.10.006).
  • Please, consider including recent evidences describing Gal-3 inhibition in COVID-19.

Author Response

RESPONSE TO REVIEWERS

The authors thank the reviewers and the editor for their precise and valuable suggestions. We have modified the manuscript accordingly and have responded to all criticisms raised. We hope that now you will find it suitable for publication.

Reviewers' comments:

REV#2: This review by de Freitas et al highlights the role of Galectin-3 inhibition in human health. The review is well-written, and written and reports interesting information through a comprehensive style.

I do not have major comments; they are mainly detailed oriented to help the authors to improve the paper.

Answer: We thanks the reviewer for the suggestions.

  • I would suggest changing the title to facilitate the comprehension for the readers.

Answer: We changed the title as suggested.

  • It could be interesting to reorganize some information.

For example, for MCP.

  • RENAL: MCP, by blocking Gal-3, is able to exert beneficial effects against renal fibrosis also in cisplatin-induced nephrotoxicity (doi: 10.1042/BSR20181803), hypertension or hypertension or acute kidney injury (doi: 10.1016/j.jacbts.2019.06.005).
  • MCP improved cardiovascular alterations in ischemia-reperfusion (doi: 10.1038/s41598-019-46119-6), ischemic heart failure (DOI: 10.3389/fphys.2019.00267), obesity or pressure overload(DOI: 10.3390/ijms18081664). Moreover, the effect of Galectin-3 inhibition with MCP in hypertension has been recently published (doi: 10.1016/j.jacbts.2020.10.006).

Answer: We reorganized the information as suggested (lines 419-475).

  • Please, consider including recent evidences describing Gal-3 inhibition in COVID-19.

Answer: We have included some information about gal-3 and COVID-19 (lines 475-490).

Reviewer 3 Report

Galectin-3 is one of the key signaling molecule playing important role in mammalian organism. This review focused on pectins as potential inhibitors of the Gal-3-mediated processes like tumor cell proliferation and growth, an inflammation as well as fibrosis etc. Information about composition of pectins (type and degree of ramification, additional modification of main and side chains, etc.), their influence on biological effects and structural features of binding sites of galectin itself were accurately collected and summarized. Most of information is devoted to experiments in vitro and this is a principle moment, because in vivo situation can be different - pectins are destroyed by microflora of the gastrointestinal tract and their effect already becomes indirect, about which authors only modest mention. Authors have tried to divide existing information onto 2 parts - in vitro and in vivo, but only in Supplementary. It should be accented in the main text as well. A chapter about products of pectin degradation and possible ways of their penetration through the gastrointestinal epithelium has to be added for the better understanding the in vivo situation. Chapter 4 should be re-written and expanded up to a broad and critical discussion of the real role of pectins for human health through the possible effect of pectins or their degradation products on Gal-3 namely in vivo. The possible application of the results obtained in vitro for in vivo studies should be discussed too.

Chapter 3 should be written from the position of galectins, not pectins. Moreover, it should be clarified from the beginning the source of galectin-3 – human or another mammals in each mentioned case. The information about galectin-3 specificity has to be extended– it is not restricted only b-Gal (for example, doi: 10.1134/S0006297915070056 and doi: 10.1007/s00418-020-01859-9). Add also the discussion about importance of multimeric presentation of  ligands for galectins  (a lot of paper are exist in this field). This can explain “abnormal” reactivity with (Araf)n, for example.

After offered improvements, it will be the good review, without analogs, which will be interesting for scientists in different fields.

Minor revisions:

There is a lot of abbreviations in the text, sometimes their presence are not justified.

line 18. Galectin-3 can not be a target of pectin. Please, re-phrase this sentence.

Fig 1A – please make it more readable. Decrypt symbols on picture B

Figure 2 – please give an description of presence of the Gal-3 in the different forms. This figure requires better description in the text.

line145 – clarify p or f in the different forms of the monosaccharides. p and f should to be marked by italic.

line 205 – it should be divided a usage of the word "sources" – sources of a pectin or literature sources. Please, clarify

line 205-216 – should be re-organized (divided, re-phrased, re-organized), because in this chapter you talk about the galectin first, you should start form galectin – not pectins.

line 218 and elswere – please check that all abbreviations are decrypted.

line 247-255 – please, correct this sentence. It’s too long and very difficult to read and understand. Use the hyphen after poly – “poly- and oligosaccharide…”

line 263-264 and elswhere – inhibition of binding of Gal-3 with what (proteins, cells?)? Please, add clarifying information. It can be important for understanding, because Gal-3 can have opposite effect in some cases.

line 266-267 – re-phrase this sentence. Interaction with what and where?

line 281 – use the italic font for “in vitro”

line 408-414 – In what system it was investigated? In vivo? In vitro?

line 474-495 – this should be a part of critical conclusion about indirect influence of pectins in vivo

line 582-584 – Figure 5. Give a decryption of the used abbreviations. There are a lot of information in the figure description which is absent in the text, for example, about paracellular pore transport of pectins or (which is more realistic) their fragments.

Table S1 – information about used experimental system (solid-phase assay, cell assays, etc.) should be added.

Author Response

RESPONSE TO REVIEWERS

The authors thank the reviewers and the editor for their precise and valuable suggestions. We have modified the manuscript accordingly and have responded to all criticisms raised. We hope that now you will find it suitable for publication.

Reviewers' comments:

REV#3: Galectin-3 is one of the key signaling molecule playing important role in mammalian organism. This review focused on pectins as potential inhibitors of the Gal-3-mediated processes like tumor cell proliferation and growth, an inflammation as well as fibrosis etc. Information about composition of pectins (type and degree of ramification, additional modification of main and side chains, etc.), their influence on biological effects and structural features of binding sites of galectin itself were accurately collected and summarized.

Answer: We thanks the reviewer for the suggestions.

Most of information is devoted to experiments in vitro and this is a principle moment, because in vivo situation can be different - pectins are destroyed by microflora of the gastrointestinal tract and their effect already becomes indirect, about which authors only modest mention.

Answer: We thanks the reviewer for the suggestions. We have now included more information to make the revision more robust in Sections 5 and 6 (lines 557-701).

Authors have tried to divide existing information onto 2 parts - in vitro and in vivo, but only in Supplementary. It should be accented in the main text as well. A chapter about products of pectin degradation and possible ways of their penetration through the gastrointestinal epithelium has to be added for the better understanding the in vivo situation.

Answer: We have highlighted throughout the manuscript if the discussed articles are derived from in vitroand/or in vivo experiments. One should note that since we did not divide the sections exclusively in in vitroand in vivo experiments, we have now clearly indicated the type of experiment discussed. We did not include a new chapter but included two paragraphs explaining the possible mechanisms of pectin and its fragments absorption, despite much more experiments should be done (and this statement was also included) (lines 491-524).

Chapter 4 should be re-written and expanded up to a broad and critical discussion of the real role of pectins for human health through the possible effect of pectins or their degradation products on Gal-3 namely in vivo. The possible application of the results obtained in vitro for in vivo studies should be discussed too.

Answer: We have re-written the whole section and included more information in Section 4 (lines 491-556).

Chapter 3 should be written from the position of galectins, not pectins. Moreover, it should be clarified from the beginning the source of galectin-3 – human or another mammals in each mentioned case. The information about galectin-3 specificity has to be extended– it is not restricted only b-Gal (for example, doi: 10.1134/S0006297915070056 and doi: 10.1007/s00418-020-01859-9). Add also the discussion about importance of multimeric presentation of  ligands for galectins  (a lot of paper are exist in this field). This can explain “abnormal” reactivity with (Araf)n, for example.

Answer: We have thoroughly changed Section 3 to make galectins as title-role. We have also updated the section with several information as suggested by the reviewer (lines 208-221, 233-245, 253-267, 270-280, 333-343).

After offered improvements, it will be the good review, without analogs, which will be interesting for scientists in different fields.

Minor revisions:

There is a lot of abbreviations in the text, sometimes their presence are not justified.

Answer: As requested we changed the text.

line 18. Galectin-3 cannot be a target of pectin. Please, re-phrase this sentence.

Answer: As remarked, it was changed.

Fig 1A – please make it more readable. Decrypt symbols on picture B

Answer: We have modified the figure.

Figure 2 – please give an description of presence of the Gal-3 in the different forms. This figure requires better description in the text.

Answer: We have modified the figure legend.

line145 – clarify p or f in the different forms of the monosaccharides. p and f should to be marked by italic.

Answer: We changed the text (lines 141-146).

line 205 – it should be divided a usage of the word "sources" – sources of a pectin or literature sources. Please, clarify

Answer: As previously suggested, the text was updated and revised, so this phrase is not in the new version.

line 205-216 – should be re-organized (divided, re-phrased, re-organized), because in this chapter you talk about the galectin first, you should start form galectin – not pectins.

Answer: We have modified the entire section to make galectins as title-role.

line 218 and elsewhere – please check that all abbreviations are decrypted.

Answer: It was performed as suggested.

line 247-255 – please, correct this sentence. It’s too long and very difficult to read and understand. Use the hyphen after poly – “poly- and oligosaccharide…”

Answer: The text was changed (lines 237-245).

line 263-264 and elsewhere – inhibition of binding of Gal-3 with what (proteins, cells?)? Please, add clarifying information. It can be important for understanding, because Gal-3 can have opposite effect in some cases.

Answer: Per clarification we have modified the text.

line 266-267 – re-phrase this sentence. Interaction with what and where?

Answer: We changed the text (lines 276-280).

line 281 – use the italic font for “in vitro”

Answer: We have changed the text.

line 408-414 – In what system it was investigated? In vivo? In vitro?

Answer: Please see new paragraph. We changed the text to clarify the investigation (lines 460-475).

line 474-495 – this should be a part of critical conclusion about indirect influence of pectins in vivo

Answer: As suggested we understand the whole paragraph is now closing the section on the indirect effects of pectins in vivo (Section 5), which is reinforcing the presentation of Table 2 that depict some experiments regarding pectin and short-chain fatty acids and by the Figure 6 (lines 578-599).

line 582-584 – Figure 5. Give a decryption of the used abbreviations. There are a lot of information in the figure description, which is absent in the text, for example, about paracellular pore transport of pectins or (which is more realistic) their fragments.

Answer: The Figure 6 in the revised version was updated and more information is now provided.

Table S1 – information about used experimental system (solid-phase assay, cell assays, etc.) should be added.

Answer: The supplementary tables were upgraded to Tables 1 and 2 and they were updated to include more information about the experimental systems.

Reviewer 4 Report

This reviews aims to assess the mechanism of action of various pectin fractions with claimed medical benefits, with a focus on interaction with galectin-3.  The authors should be commended for having included a part (4) on the fact that the role of galectin-3 interaction for pectin mechanism of action has been challenged. They also include parts on galectin-3 independent activities of pectins (5) and  a section asking what should be considered main mechanism of action (6). These are excellent to include and are the main value of this review compared to other on pectins and galectin-3. Many previous reviews and original papers on pectin derived saccharides and galectin-3 have been plagued by a predetermined and wishful thinking that galectin-3 shall be a mechanism of action, on part of the authors, which has prevented critical evaluation of the experimental data.

So this paper would be further improved by some examples of critical evaluation. Two examples are given here:

For the paper by Hu et al, 2020 in Table 2 it is said that the “Observed biological effect” is “Reduction of inducted inflammatory and oxidative stress through Gal-3.” So what is the evidence for this?  The authors treat cultures of islet cells with streptozotocin (STZ) or some cytokines for 24 hours and get various deleterious effect (decreased viability, increased apoptosis and ROS production etc.). If they pre and co-incubate with some pectin fractions (2 mg/ml) the cells appear to be rescued form some of the deleterious effects. To find evidence for a role of Gal-3 they included 20 mM lactose in the preincubation (1 hour) together with the pectin and now found theta the pectin did not rescue the cells after 24 hours. The problem is, this does not show in any way that binding of the pectin to Gal-3 should be its mechanism of action. The lactose could have acted in any number of ways, besides binding Gal-3, including binding other galectins, but also having had completely different effects.  Incubating a cell with 2 mg/ml of a polysaccharide is also bound to have a variety of effects. Is the polysaccharide endocytosed. Does it kill cells, and the author only analyzed the surviving ones (how well controlled is the paper). So the statement under “Observed biological effect” for this paper is false.

 For the paper by Xue et al, 2019 in Table 2 it is said that the “Observed biological effect” is “Selective inhibition of Gal-3 and ROS/ERK pathway, promotion of T-cell proliferation and IL-2 expression.” In this paper the authors first study cultured cells (Jurkat) and added Gal-3 for up to 18 hours. This gave increased cell activation (CD69) and IL-2 secretion and apoptosis among other effects. Then the author show that different pectin fractions (2 mg/ml) have prevent these effects to different degree (interesting by itself). However, they also show, when analyzing signaling pathways that other inhibitors (for these pathways) can also inhibit the effects. Moreover the experimental times 18 hours, under which lots of things could have happened. So where is the evidence that the pectins have their action by binding galectin-3?  In the second part of this paper, the authors treat tumour bearing mice by ip injection of pectins but there are no data on galectin-3, except that its serum level is higher with tumors.  So the statement under “Observed biological effect” is again misleading.

As these examples show, the authors of a good review need to read referred papers in detail, and evaluate them with a fresh unbiased mind, assuming nothing about galectin-3.

As general comments, the paper is not very clear and well written, and piles up lots of statements after each other with many reference, but not so much logic and thought along the way.

Figures are not well designed with too small text in some and too little explanation, especially Fig. 2 and 5. Why do they show intestine in particular?

In Table 1 and 2 the references do not have number, so hard to find in refertence list, which is not alphabetical. Needs to be fixed.

In all a not well prepared manuscript. BUT the inclusion of section 4 -6, a first among reviews on pectins relationships to galectin-3, makes it worthwhile.

Author Response

RESPONSE TO REVIEWERS

The authors thank the reviewers and the editor for their precise and valuable suggestions. We have modified the manuscript accordingly and have responded to all criticisms raised. We hope that now you will find it suitable for publication.

Reviewers' comments:

REV#4: This reviews aims to assess the mechanism of action of various pectin fractions with claimed medical benefits, with a focus on interaction with galectin-3.  The authors should be commended for having included a part (4) on the fact that the role of galectin-3 interaction for pectin mechanism of action has been challenged. They also include parts on galectin-3 independent activities of pectins (5) and  a section asking what should be considered main mechanism of action (6). These are excellent to include and are the main value of this review compared to other on pectins and galectin-3. Many previous reviews and original papers on pectin derived saccharides and galectin-3 have been plagued by a predetermined and wishful thinking that galectin-3 shall be a mechanism of action, on part of the authors, which has prevented critical evaluation of the experimental data.

Answer: We thanks the reviewer for the suggestions.

So this paper would be further improved by some examples of critical evaluation. Two examples are given here:

For the paper by Hu et al, 2020 in Table 2 it is said that the “Observed biological effect” is “Reduction of inducted inflammatory and oxidative stress through Gal-3.” So what is the evidence for this?  The authors treat cultures of islet cells with streptozotocin (STZ) or some cytokines for 24 hours and get various deleterious effect (decreased viability, increased apoptosis and ROS production etc.). If they pre and co-incubate with some pectin fractions (2 mg/ml) the cells appear to be rescued form some of the deleterious effects. To find evidence for a role of Gal-3 they included 20 mM lactose in the preincubation (1 hour) together with the pectin and now found theta the pectin did not rescue the cells after 24 hours. The problem is, this does not show in any way that binding of the pectin to Gal-3 should be its mechanism of action. The lactose could have acted in any number of ways, besides binding Gal-3, including binding other galectins, but also having had completely different effects.  Incubating a cell with 2 mg/ml of a polysaccharide is also bound to have a variety of effects. Is the polysaccharide endocytosed. Does it kill cells, and the author only analyzed the surviving ones (how well controlled is the paper). So the statement under “Observed biological effect” for this paper is false.

Answer: We thanks the reviewer for this minutely consideration. We have changed the statement regarding this reference (lines 297-299), but we have also changed the title of Table 2 column to clarify the situation when the “Observed experimental effects” cannot be stated as a truly “biological effect”. Moreover, the column “Observed experimental effects” was updated to be with more information.

For the paper by Xue et al, 2019 in Table 2 it is said that the “Observed biological effect” is “Selective inhibition of Gal-3 and ROS/ERK pathway, promotion of T-cell proliferation and IL-2 expression.” In this paper the authors first study cultured cells (Jurkat) and added Gal-3 for up to 18 hours. This gave increased cell activation (CD69) and IL-2 secretion and apoptosis among other effects. Then the author show that different pectin fractions (2 mg/ml) have prevent these effects to different degree (interesting by itself). However, they also show, when analyzing signaling pathways that other inhibitors (for these pathways) can also inhibit the effects. Moreover the experimental times 18 hours, under which lots of things could have happened. So where is the evidence that the pectins have their action by binding galectin-3?  In the second part of this paper, the authors treat tumour bearing mice by ip injection of pectins but there are no data on galectin-3, except that its serum level is higher with tumors.  So the statement under “Observed biological effect” is again misleading.

Answer: We thanks the reviewer for this carefully consideration. We have changed the statement regarding this reference (lines 256-258) and modified the title of Table 2 column. Further, it should be noted that the description of this article in the “Observed experimental effects” has not addressed the Gal-3 inhibition (Down-regulation of Gal-3, TLR and MyD88, decreased expression of IL-1β, IL-18 and TNF-α), and is now clarified.

As these examples show, the authors of a good review need to read referred papers in detail, and evaluate them with a fresh unbiased mind, assuming nothing about galectin-3.

As general comments, the paper is not very clear and well written, and piles up lots of statements after each other with many reference, but not so much logic and thought along the way.

Answer: We thanks the reviewer for the suggestions. The manuscript was revised to improve the logic presentation of ideas.

Figures are not well designed with too small text in some and too little explanation, especially Fig. 2 and 5. Why do they show intestine in particular?

Answer: The figures are now better. The intestinal model was chosen as it is simple to explain the interface between endogenous and exogenous stimuli that differently activate Gal-3 functions. Of note, the probable effects of pectin throughout the gastrointestinal tract. Pectin and its fragments may be considered as bioactive compounds when ingested (as food components or as dietary supplements), we explored this issue. Nevertheless, some experiments were done through other routes of administration, so this was made clearer in the new version of the manuscript, please see lines 423-428.

In Table 1 and 2 the references do not have number, so hard to find in refertence list, which is not alphabetical. Needs to be fixed.

Answer: As requested we have now included the references numbers in Tables 1 and 2 for clarity.

In all a not well prepared manuscript. BUT the inclusion of section 4 -6, a first among reviews on pectins relationships to galectin-3, makes it worthwhile.

Answer: We trust that hop we have addressed all the reviewers’ criticisms and the new version could reach the reviewer’s expectation to be considered for publication, since in all sections were included more information and several text reorganization were done.

Round 2

Reviewer 3 Report

Change a little the Abstract - add the information about performed critical analysis of in vitro and in vivo data for the Gal-3 inhibition by pectins.

It looks like there are several repeating sentences connecting to the digestion of a pectin as dietary fibers in the text. Please, check.

Figure 3. Please, give the pictures of Gal-3 in the different forms directly on the Figure and give a description (pentameric and monomeric Gal3 look like immunoglobulins - please, explain in the Figure that these are Gal-3)

Line 270-272  - this sentence should be used as a supposition why Gal-3 can bind and inhibited by non-beta-Gal fragments of pectins.

Author Response

Article Reference: #1532325

Journal: Biomolecules

Title: The Complex Biological Effects of Pectin: Galectin-3 Targeting as Potential Human Health Improvement?

RESPONSE TO REVIEWERS

The authors thank the reviewers for their suggestions. We have modified the manuscript accordingly and have responded to all appreciations. We hope that now you will find it suitable for publication.

Reviewers' comments:

Rev#3.

Change a little the Abstract - add the information about performed critical analysis of in vitro and in vivo data for the Gal-3 inhibition by pectins.

Answer: We thoroughly revised the Abstract.

It looks like there are several repeating sentences connecting to the digestion of a pectin as dietary fibers in the text. Please, check.

Answer: We checked the text and changed to better clarify this point.

Figure 3. Please, give the pictures of Gal-3 in the different forms directly on the Figure and give a description (pentameric and monomeric Gal3 look like immunoglobulins - please, explain in the Figure that these are Gal-3)

Answer: Figure 3 was changed as suggested.

Line 270-272  - this sentence should be used as a supposition why Gal-3 can bind and inhibited by non-beta-Gal fragments of pectins.

Answer: We changed the sentence accordingly (lines 273-277).

Reviewer 4 Report

The paper is improved but still uncritical. It is valuable with such an extensive reference list and Discussion also of controversies. But the tone in the title and abstract is still that the authors believe and readers shall expect that the various pectin and other plant polysaccharides act as galectin-3 inhibitors. Even of many publications claim that even in their titles, the actual evidence is not there. One example is given below.  

So the title is OK but should be ended with a question mark

In the Abstract is written: “The discussion highlighted by this review demonstrates that pectins are promissory food-derived molecules for bioactive functions, such as galectin-3 inhibition and its consequent beneficial effects to humans”.

The first part may be true, but the second part about galectin-3 is not.

So the Abstract should contain a statement like: “Pectin fractions have been proposed to act biologically as inhibitors of galectoin-3, but this is far from clear or experimentally proven. This review tries to evaluate the evidence for and against.“

One example picked at random is Ref 106. It has the title: Galectin- blockade reduces renal fibrosis … etc.
In the paper rats are treated to induce renal fibrosis and various molecules measured, including galectin-3, go up. Then the rats are given MCP in drinking water and the fibrosis is ameliorated and the various molecule go down in levels. Where is the evidence for galectin-3 blockade? How does the author envisage that this works from MCP in the drinking water via intestine to the kidney. There is nothing about this problem in the paper. The authors calls it “pharmacological inhibition of Gal-3 with MCP”, which is nonsense.

This is just one example, but the literature on pectin fractions and galectin-3 is full of similar cases. So, readers need to be warned to not trust Titles or Abstracts, but instead read the actual data and make their own thoughtful conclusions.

A small point on basic facts about galectins and references. On line 216 – 220 the authors introduce the subsites ABCDE on the S-face of the galectin CRD, which is fine. But naming them in terms of dimers as is done is wrong. They are not dimers or tetramers or anything like that. They are subsites, or just a way to name reference points on the galectin-3 CRD to make talking about binding activities easier. This way of describing the CRD bining site was first introduced in Leffler et al.  Introduction to galectins  Glycoconj J . 2002;19(7-9):433-40.

But a bit old now, so a more modern reference for basic galectin knowledge could be:

Johannes et al. Galectins at a glance J Cell Sci 2018 131(9):jcs208884.

doi: 10.1242/jcs.208884.

Author Response

Article Reference: #1532325

Journal: Biomolecules

Title: The Complex Biological Effects of Pectin: Galectin-3 Targeting as Potential Human Health Improvement?

RESPONSE TO REVIEWERS

The authors thank the reviewers for their suggestions. We have modified the manuscript accordingly and have responded to all appreciations. We hope that now you will find it suitable for publication.

Reviewers' comments:

Rev#4.

The paper is improved but still uncritical. It is valuable with such an extensive reference list and Discussion also of controversies. But the tone in the title and abstract is still that the authors believe and readers shall expect that the various pectin and other plant polysaccharides act as galectin-3 inhibitors. Even of many publications claim that even in their titles, the actual evidence is not there. One example is given below.  

Answer: We thanks the advice, and we carefully changed the title (added a question mark), the abstract and several parts of the text to increase the perception that galectin-3 inhibition by pectin is a theme far from being established and is very controversial.

So the title is OK but should be ended with a question mark

Answer: We ended the title with a question mark.

In the Abstract is written: “The discussion highlighted by this review demonstrates that pectins are promissory food-derived molecules for bioactive functions, such as galectin-3 inhibition and its consequent beneficial effects to humans”.

The first part may be true, but the second part about galectin-3 is not.

Answer: We thoroughly revised the Abstract.

So the Abstract should contain a statement like: “Pectin fractions have been proposed to act biologically as inhibitors of galectoin-3, but this is far from clear or experimentally proven. This review tries to evaluate the evidence for and against.“

Answer: We included this statement in the Abstract.

One example picked at random is Ref 106. It has the title: Galectin- blockade reduces renal fibrosis … etc.
In the paper rats are treated to induce renal fibrosis and various molecules measured, including galectin-3, go up. Then the rats are given MCP in drinking water and the fibrosis is ameliorated and the various molecule go down in levels. Where is the evidence for galectin-3 blockade? How does the author envisage that this works from MCP in the drinking water via intestine to the kidney. There is nothing about this problem in the paper. The authors calls it “pharmacological inhibition of Gal-3 with MCP”, which is nonsense.

This is just one example, but the literature on pectin fractions and galectin-3 is full of similar cases. So, readers need to be warned to not trust Titles or Abstracts, but instead read the actual data and make their own thoughtful conclusions.

Answer: We understand the point of view, so we changed the text to give the idea some authors did not experimentally explore the pectin absorption, so readers must be careful to conclude a systemic galectin-3 inhibition by pectic fractions that were supposed to be absorbed (lines 412-415, 436-438, 466-473, 506-510, 560-567).

A small point on basic facts about galectins and references. On line 216 – 220 the authors introduce the subsites ABCDE on the S-face of the galectin CRD, which is fine. But naming them in terms of dimers as is done is wrong. They are not dimers or tetramers or anything like that. They are subsites, or just a way to name reference points on the galectin-3 CRD to make talking about binding activities easier. This way of describing the CRD bining site was first introduced in Leffler et al.  Introduction to galectins  Glycoconj J . 2002;19(7-9):433-40.

But a bit old now, so a more modern reference for basic galectin knowledge could be:

Johannes et al. Galectins at a glance J Cell Sci 2018 131(9):jcs208884.

doi: 10.1242/jcs.208884.

Answer: We thank the reviewer for pointing out this mistake and the text was modified as suggested (lines 209-224, 236-248).
